# Volumetric imaging of fast cellular dynamics with deep learning enhanced bioluminescence microscopy

Luis Felipe Morales-Curiel[1], Adriana Carolina Gonzalez[1,5], Gustavo Castro-Olvera[1,5], Li-Chun (Lynn) Lin[1], Malak El-Quessny[1], Montserrat Porta-de-la-Riva[1], Jacqueline Severino[2], Laura Battle Morera[2], Valeria Venturini[2,3], Verena Ruprecht[2,3,4], Diego Ramallo[1], Pablo Loza-Alvarez[1] & Michael Krieg[1✉]

Bioluminescence microscopy is an appealing alternative to fluorescence microscopy, because it does not depend on external illumination, and consequently does neither produce spurious background autofluorescence, nor perturb intrinsically photosensitive processes in living cells and animals. The low photon emission of known luciferases, however, demands long exposure times that are prohibitive for imaging fast biological dynamics. To increase the versatility of bioluminescence microscopy, we present an improved low-light microscope in combination with deep learning methods to image extremely photon-starved samples enabling subsecond exposures for timelapse and volumetric imaging. We apply our method to image subcellular dynamics in mouse embryonic stem cells, epithelial morphology during zebrafish development, and DAF-16 FoxO transcription factor shuttling from the cytoplasm to the nucleus under external stress. Finally, we concatenate neural networks for denoising and light-field deconvolution to resolve intracellular calcium dynamics in three dimensions of freely moving *Caenorhabditis elegans*.

[1] ICFO, Institut de Ciencies Fotòniques, Castelldefels, Spain. [2] Center for Genomic Regulation (CRG), The Barcelona Institute of Science and Technology, Barcelona, Spain. [3] Universitat Pompeu Fabra (UPF), Barcelona, Spain. [4] ICREA, Pg. Lluis Companys 23, 08010 Barcelona, Spain. [5]These authors contributed equally: Adriana Gonzalez, Gustavo Castro-Olvera. ✉email: michael.krieg@icfo.eu

Fluorescence microscopy has enabled unprecedented discoveries and became the major imaging modality in molecular and cellular bioscience. However, strong autofluorescence of the native tissues[1] often obscures and blurs the signal from specific labels in biological samples[2], while intrinsic photosensitivity of cells and animals, such as *Caenorhabditis elegans*[3], planaria[4], or mouse preimplantation embryos[5], interferes with imaging experiments that require an excitation light source. In addition, the incompatibility in excitation and emission profiles of some fluorescent proteins can make it difficult to simultaneously image multiple fluorophores, e.g., CFP and GFP cannot be imaged at the same time with a single filtercube, fluorescence imaging is also limited if the excitation spectrum of the chromophore overlaps with that of a photosensitizer (e.g., 470 nm for Channelrhodopsin[6] and Tulips[7]). Further, the high excitation intensities that are necessary to obtain fluorescent images with extreme photon-starved samples render fluorescence microscopy potentially phototoxic and limit the lifetime of the fluorescent probe[8]. These drawbacks can be overcome by implementing bioluminescent probes as contrast labels, since they can be genetically encoded to tag any protein of interest, and do not need an external excitation light source.

However, several factors have limited bioluminescent microscopy as a mainstream technique in a cell biological or pharmacological laboratory. First, bioluminescent enzymes require a chemical co-factor, e.g., luciferin, coelenterazine, or furimazine and their derivatives, as a photon source, which becomes oxidized prior to photon emission. Many of these cofactors are poorly soluble in water, have poor membrane permeability which reduces diffusion across cells thus lowering the bioavailability inside tissues and animals[9,10]. In addition, luciferases have a slow catalytic turnover[11], determined by enzyme affinity for the substrate, or the quantum yield which is defined as the probability of emitting a photon per enzymatic reaction[12]. The optimization of these cofactors and the enzymes through chemical and bioengineering is an active research field and have greatly improved the versatility and photon emission characteristics. Engineered luciferases based on deep sea shrimp, termed Nanolanterns[13,14], in which the luciferase moiety is fused to a fluorescent protein, have among the highest quantum yield and catalytic turnover and bear the potential to select the emission wavelength[12]. Thus, a whole spectrum of light-emitting proteins can be tailored to a specific need[15]. Even with these optimizations and cofactors, at saturated conditions in vitro where chemical delivery is not limiting, these enzymes only produce $\approx 10$ photons per second, as compared to 3000 photons per millisecond for conventional fluorescence microscopy[16], which greatly limits the number of emitted photons per time interval and concomitant signal-to-noise ratio (SNR). Thus, to obtain high-SNR images calls for long exposure times in the seconds or even tens of seconds scale, even with state-of-the-art EMCCD cameras, which is incompatible with fast biological dynamics. Even when the sample is immobile, long exposure times greatly limit the image acquisition and experimental throughput, e.g., for pharmacological studies and biosensor application during drug screening and cancer research[17]. As a consequence, the resultant images are often noisy, with low resolution and limited to static images without temporal and three-dimensional resolution. Frequently, one wants to post-process these noisy images to obtain a clear pictures at optimal resolution; however, because noise, and many image features such as edges and texture share high frequency components, procedures that reduce noise often introduce artefacts. In general, the image after a denoising operation should contain smooth flat areas, unblurred edges while preserving resolution and avoid the generation of artefacts. Thus, recovering meaningful information is mathematically complex and solutions are not unique[18].

Several methods have been deployed previously to increase sensitivity and reduce noise in ultra-photon-starved images. Early attempts to clean bioluminescent images through blind deconvolution produced modest improvements[19,20].

Convolutional neural networks (CNN) have been shown to outperform classical algorithms in different tasks including denoising[21–23], 3D reconstruction methods[24–26], and super-resolution imaging[27–29]. In order to leverage the machine learning models like neural networks, it is necessary to build a training pipeline and an inference pipeline. For supervised learning (SL) workflows, the training pipeline includes steps to preprocess the data, train models on input data and their corresponding target data, and validate their results. The inference pipeline uses the optimal model from the training pipeline to generate predictions on unseen input data.

The goal of a denoising model is to obtain a high-SNR image from a low-SNR image without losing image feature and resolution. This requires training a model on degraded images $x^i$ and their corresponding high-quality target images $y^i$, which are usually acquired using fluorescence microscopy. A popular approach is content-aware image restoration (CARE), which takes advantage of the knowledge of the underlying sample structure. In microscopy, these are usually clean model images, acquired at imaging condition that would result in sharp and low noise images without necessarily preserving physiological activity (e.g., fixed and stained cells). CARE consists of a set of U-Net type architectures to perform common restoration tasks, such as image denoising, surface projection and restoration of isotropic resolution using supervised learning. It has been able to achieve impressive results with extremely photon-starved fluorescent samples, such as images derived from intrinsically photo-sensitive planaria[21]. CARE is compatible with image acquisition at low laser power for fluorescence microscopy, which is important for light-sensitive specimen and to reduce photo-toxicity. By extension, CARE pipelines are also uniquely poised to overcome noise problems in bioluminescence microscopy.

Due to the long exposure times necessary to acquire high-SNR images with current setups designed for bioluminescent microscopy, and the risk of sample movement, three-dimensional bioluminescence imaging has remained elusive. Light Field microscopy (LFM) has become the standard imaging choice for instantaneous volumetric imaging[30] using a single exposure, and thus promises to reduce the volumetric imaging rate. Instead of acquiring a volume with $n$ steps, only a single exposure is needed to capture the whole 3D representation of the scene. Before any useful information can be extracted from a raw 2D light field (LF) image, it needs to be deconvolved[30]. However, the analytical deconvolution based on a known point-spread function is very time-consuming and requires high amounts of computational resources due to the iterative nature of the algorithms to perform the light field deconvolution[31,32]. With the aim to reduce the temporal cost in the deconvolution process, several CNN models were proposed[24–26]. LFMNet reconstructs LF images trained with experimental light field images and their corresponding confocal scans[26]. However, since the GT and the input must be aligned due to the wise-pixel correlation during the supervised training, LF images and the corresponding confocal stacks must be registered before the training takes place. Therefore, for proper registration, the method required a light field deconvolution procedure to prepare the training dataset which, as mentioned above, is time consuming and prone to artifacts. Often, deep-learning models memorize instrument-specific features of the images they have been trained, which limits their transferability. To generalize the performance of the models on various microscopy setups, HyLFM-Net[25] leverages a transfer-learning approach to continuously verify and validate the accuracy in the reconstruction

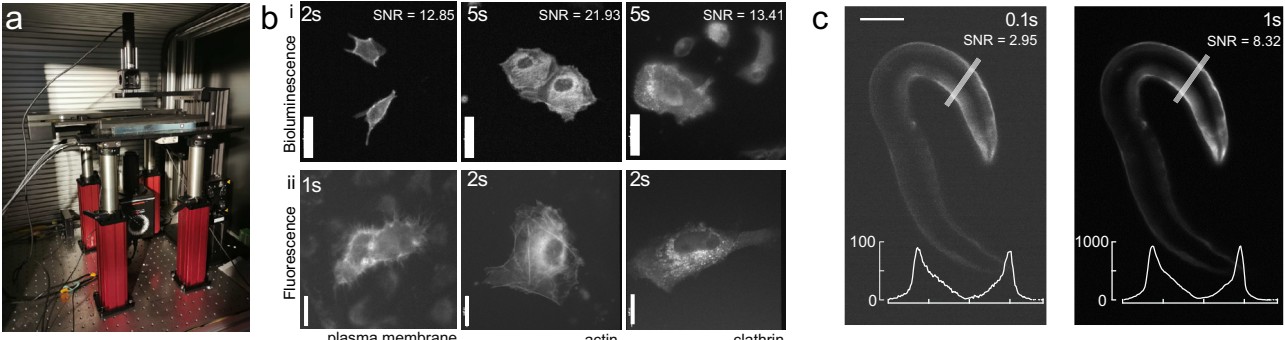

**Fig. 1 Optimized bioluminescence microscopy. a** Photograph of the optimized LowLiteScope. **b** Bioluminescent (i, ×20 magnification) and epifluorescent (ii, ×63 magnification) images of cell expressing the indicated marker taken on the LowLiteScope (i) or a commercial epifluorescence (ii) microscope, respectively. Exposure times indicated in the top left of each image. Scale bar = 20 µm. **c** Bioluminescent images of an immobilized worm expressing a turquoise-enhanced Nanolantern (TeNL) in the body wall muscle at different exposure times. Insets show the intensity profile across the line indicated in the image. Scale bar = 50 µm.

performance of high-resolution LF images and ground-truth images acquired on different microscopy setups. Transfer-learning approaches are particular useful to generalize specific deep-learning model for wider applicability and especially in cases where training data are difficult and expensive to collect. Wang et al.[24] introduced a view channel depth (VCD) neural network to overcome the non-uniform angular resolution and avoid the reconstruction of artifacts in the network inference. They generated synthetic light field data by performing a forward projection between 3D high-resolution images and a simulated light field PSF of the system. Trained networks were able to perform calcium imaging of *C. elegans* and predicted the blood flow movement in the heart of a zebrafish. This technique has proven to be superior due to its easy implementation on account of relying solely on synthetic data and its independence from other imaging techniques.

Here, we overcome several prohibitive challenges, and demonstrate the use of bioluminescence as an imaging modality in the millisecond range. We constructed a new microscope with a shortened, optimized optical path, light field detection and single photon resolution in combination with machine learning to maximally reduce camera exposure time. Because accurate inference requires high-quality training data of biological samples, we built training and inference pipelines for our deep-learning models using two concatenated neural networks and a transfer-learning approach with the aim to increase the signal-noise ratio and reconstruct four dimensional information from a time series of 2D images. We demonstrate this approach to image nuclear dynamics in mouse embryonic stem cells, 3D imaging of zebrafish epithelial tissues and whole-body calcium imaging in muscles of freely moving *Caenorhabditis elegans*. Thus, bioluminescence microscopy could mature into an essential tool for synthetic biology[33], in approaches that harness endogenous light generation with luciferases that are not limited to microscopy approaches but also functional bioluminescence optogenetics[34–37].

## Results
**LowLiteScope.** Due to the low quantum yield of luciferases, standard optical microscopes are poorly suitable to produce bioluminescent images and dedicated instruments are commonly used[38]. Indeed, we only observed a weak signal from Nano-lanterns transfected into HeLa cells or transgenic *C. elegans* on a commercial compound microscope at the maximum exposure time of our camera (Supplementary Fig. 1). To increase the photon collection efficiency, we thus conceived a microscope with

an ultra-compact optical axis, and with a single photon-resolving, quantitative, qCMOS camera (Fig. 1a). With this new setup, we were able to obtain high-SNR images for cells transfected with Nanolantern fusions to clathrin, actin and the plasma membrane marker lyn[14] and supplemented with the co-factor Hikarazine (see Methods[39]) for exposure times down to 2s (Fig. 1b, i). Importantly, even without any further treatment, these images were comparable to fluorescence images acquired at a similar exposure time at higher magnification (Fig. 1b, ii) on a conventional, epifluorescence microscope. As expected, no auto-fluorescence was observed in the luminescence images due to the absence of an external excitation light source, in contrast to the fluorescence images (Fig. 1b). We next established that the optimized bioluminescence imaging protocol enhances the photon collection efficiency in living animals. We created a transgenic *C. elegans* line that expresses a turquoise Nanolantern in their body wall muscles[34] and immobilized individual animals for imaging on an agar pad in presence of the luciferin. In agreement with our results from tissue culture cells, we observed a strong specific signal at longer exposure time and even for exposure times down to 50 ms (Fig. 1c). Taken together, these technical improvements dramatically augmented the quantity of photons detected allowing us to reduce the exposure time without additional post-processing. Capturing the ability to record ultra-photon-starved samples, we refer to our setup as 'LowLiteScope'.

**Transcription factor dynamics under external stress.** Photobleaching during fluorescence microscopy is a well-known indicator for potential photo damage to the cell[8], especially at lower wavelengths commonly used for one photon excitation live-cell imaging. Without the requirement of an excitation light source, bioluminescence has the advantage to circumvent potentially phototoxic effects[40]. To understand to what extent bioluminescent and fluorescent excitation induces a cellular stress response, we generated transgenic animals expressing mNeonGreen-NanoLantern fused to DAF-16 (Fig. 2a), the *C. elegans* ortholog of FoxO transcription factors which promotes longevity[41] and acts as a reporter for various stresses, including reactive oxygen species[42] but also heat[43]. Importantly, this stress reporter can be detected by direct excitation of the mNG using fluorescence microscopy, or independent of external excitation light through the bioluminescence of its NanoLantern. In its inactive state, DAF-16 resides in the cytoplasm, but translocates rapidly into the nucleus, where it induces the transcription of stress protective genes, when the animal is exposed to cytotoxic insults[43] (Fig. 2a). We first verified that the indicator is functional and has the ability

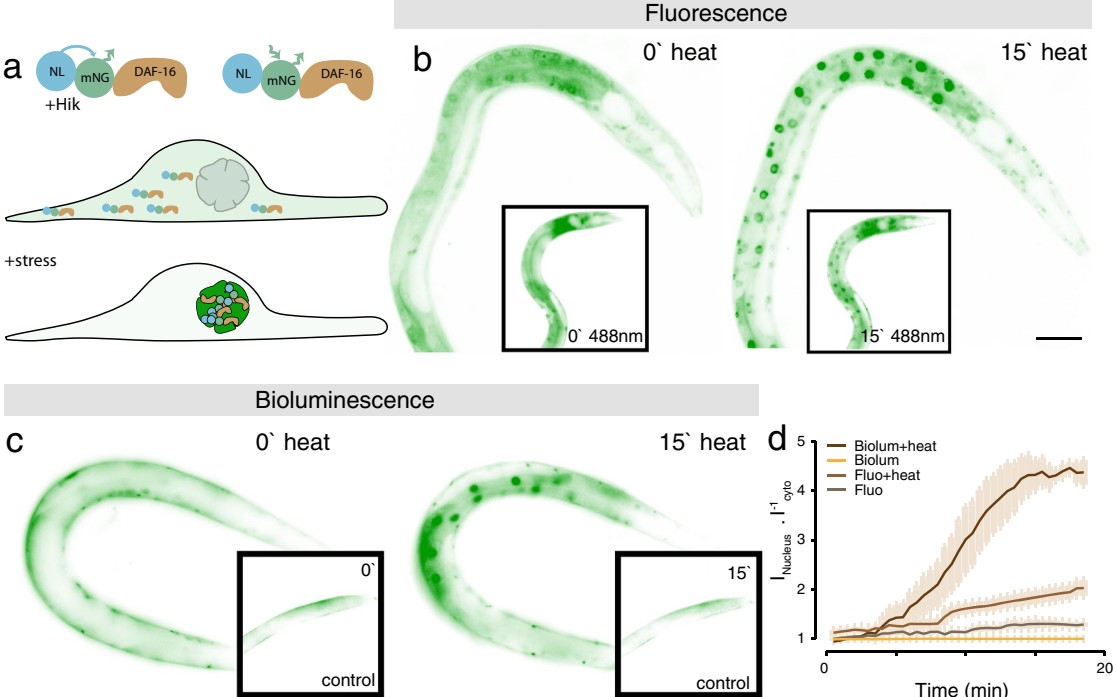

**Fig. 2 Stress reporter activation under external illumination and heat. a**, Sketch of the bioluminescent reporter. the full-length *daf-16* cDNA is fused to a mNeonGreen fluorophore and a NanoLuc luciferase, called GeNL. **b**, **c** Fluorescent (**b**) and bioluminescent (**c**) DAF-16::GeNL before (0') and after (15') exposure to 37 °C heat stress. Inset shows animals at the same timepoints without external heat stress, but continuous external illumination.
**d** Quantification of the nuclear/cytoplasmic DAF-16 ratio (Mean ± standard deviation (SD)) over 18 min of the experiment in the four tested conditions (representative for $N = 5$ animals). Scale bar = 40 μm.

to report the animal's exposure to stresses, and imaged cytoplasmic/nuclear shuttling during the application of a heat shock. As expected, DAF-16 rapidly translocated into the nucleus immediately after the onset of the heat shock independent of whether fluorescence (Fig. 2b) or bioluminescence is used (Fig. 2c; Supplementary Movie 1). Unexpectedly, the nuclear/cytoplasmic intensity ratio of the bioluminescent reporter was higher than that of the fluorescent reporter, presumably due to the absence of autofluorescent background in the bioluminescent images which provided a higher dynamic range (Fig. 2d).

We then used these transgenic animals and compared the response of DAF-16 during fluorescence and bioluminescence imaging at room temperature. Importantly, the transgenic animals that were recorded with fluorescence microscopy showed a significant increase in nuclear DAF-16 localization with time (Fig. 2b inset, f; R = 0.95, *p* < 1e-15). In contrast, DAF-16 remained largely inactive during bioluminescent microscopy (Fig. 2c, d; R = 0.56, *p* = 1e-5). We thus speculated that the nuclear translocation of DAF-16 might be activated by the reactive oxygen species that were generated during the fluorescence illumination[8,44]. Even though this effect is subtle, it is noticeable and might confound the effect of other stresses. This analysis also provided the important insight that the exposure of the animal to the luciferase co-factor does not activate DAF-16 FoxO pathways and thus is itself not cytotoxic. Taken together, the application of bioluminescent reporters offers a higher dynamic range due to the absence of background autofluorescence and could possibly guide the discovery of stress pathways that would otherwise be obscured by the cellular response to external light.

**Content-aware restoration of photon-starved bioluminescent images**. In absence of background 'signal' (bleedthrough, uneven illumination, autofluorescence) in bioluminescent samples, we

reasoned that prior knowledge of the underlying sample structure should facilitate image reconstruction of shot-noise limited bioluminescence microscopy using deep-learning-based content-aware image restoration (CARE) algorithms[21]. To clean up these images and increase the SNR, we combined these bioluminescent microscopy with a convolutional neural network model pipeline that transforms a degraded image to a desired high-quality target[21]. Because CARE models for *C. elegans* and bioluminescence are non-existent, we first developed a generalizable training pipeline to predict high quality, ground-truth images from noisy input. We thus collected image pairs derived from fluorescence microscopy acquired at extremely low exposure times reflecting the noisy input with poor SNR and used high-SNR images from long exposure times as the ground-truth target (Fig. 3a). The training dataset consisted of animals transgenic for mTurquoise expressed in body wall muscles, which showed highly specific signal that was much stronger than the autofluorescence within the region of interest which remained undetectable. To generalize and achieve a high variety of body postures and thus 2D intensity distributions, we recorded the fluorescence signal from the body wall muscles in freely moving animals. After data collection and preprocessing, we varied the hyperparameters to find the optimal network configuration[21,45] specific for our training dataset (Fig. 3b, Supplementary Fig. 2), and evaluated its out-of-sample performance on unseen noisy images using the structural similarity (SSIM) and Normalized Root Mean Squared Error (NRMSE) metrics for training quality (Fig. 3c, Supplementary Fig. 2 and Methods). We then built an inference pipeline with the model showing the highest confidence and lowest error to predict the ground truth from the noisy bioluminescent images which turned out to be completely clean and devoid of artifacts (Fig. 3d). In agreement with previous results, we found that the performance of the supervised learning procedure is superior to the self-supervised method[46].

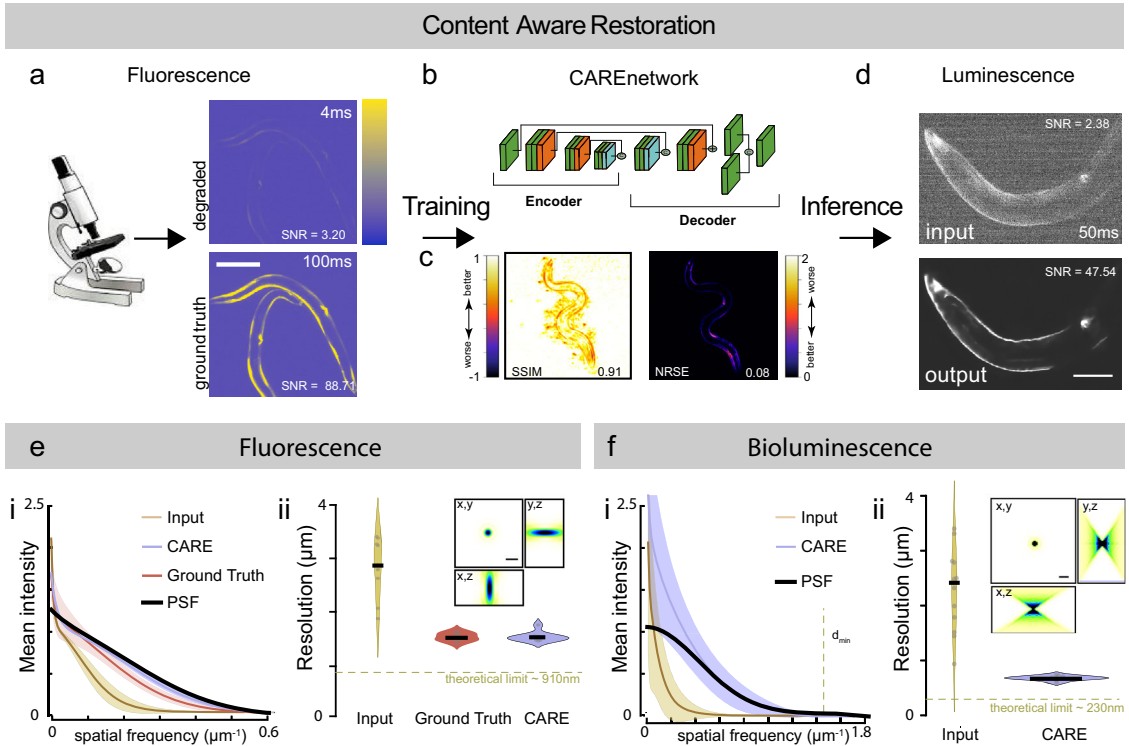

**Fig. 3 CARE training pipeline for bioluminescent images. a** Schematic of the content-aware restoration deep-learning pipeline. Pairs of images were simultaneously collected with a beamsplitter in an epifluorescent microscope at different exposure times to create a low and high-SNR training dataset. **b** After subpixel registration, a deep neural network is trained to restore the test image from the high-SNR ground truth. **c** Structural similarity (SSIM) and Normalized Root Mean Squared Error (NRMSE) of the predicted images vs the ground truth (see also Supplementary Fig. 1). **d** The trained network is then used to restore low-SNR bioluminescent images. Scale bar = 50 μm. **e, f** CARE recovers high frequency components from degraded images. Estimation of the smallest resolvable feature from the noisy fluorescence images (**e**), Ground truth and deep-learning predictions and the **f** bioluminescent images. Fluorescence images were acquired with a ×10/0.3 objective and bioluminescence images were acquired with a ×40/1.25 silicon oil objective. (i) Average azimuth intensity of the Fourier-transformed image, plotted against frequency in the 2D power spectrum image. Mean ± SD (ii) Violin plot of the smallest resolvable feature ($N = 9$ and 15 technical replicates (images) for each condition, respectively). Black line is the median of the distribution. Inset show the synthetic PSF calculated assuming 470 nm emission light using the Born& Wolf method[81]. Theoretical resolution limit was calculated according to $0.61\lambda/NA_{objective}$. Scale bar = 2 μm.

We next determined the impact of the deep-learning restoration on the spatial resolution of the images. Typically, resolution is estimated from the classical Rayleigh limit or measured directly from the intensity profile of subresolved image features, e.g., Full Width Half Maximum (FWHM). To estimate the smallest resolvable feature detectable in our images, we apply a fast Fourier transform (FFT) on the image of interest and measure the spatial frequency distribution. The largest frequencies can either be used to estimate the FWHM of the point-spread function (see refs. [47,48] and Methods) or assess the quality of the resulting images[49,50]. To estimate whether CARE improves the resolution of the noisy images, we first applied an FFT, calculated the radial average of the resulting power spectrum and fitted to a rationalized equation based on a Gaussian point-spread function to extract $d_{m}in$, the minimally resolvable distance[49]. We then compared fluorescent input images with large noise contribution to the ground-truth images used for the training process and found, not surprisingly, that the presence of noise degenerated the image resolution (Fig. 3e). We then applied the same procedure to the images derived from the CARE pipeline after denoising and were able to recover the high frequencies seen in the ground-truth images (Fig. 3e ii). With this procedure in hand, we then applied the method to the noisy bioluminescence images and their CARE predictions and found an impressive increase in resolution, only slightly lower than the theoretical value (Fig. 3f). To further characterize the performance of the deep-learning CARE

pipelines on bioluminescence image restoration, we determined if there is a need for a minimal signal/noise ratio in the input images such that these can be restored using our trained CARE pipelines. Whereas we have not encountered strict limits for the prediction, because we have successfully restored degraded images with a SNR slightly above one, the reliability of the restoration increased noticeably for input image with an SNR > 2.5. To quantify this, we plotted the SNR of the input images against the obtained SSIM that describes the quality of the restoration, and found that robust high-quality inferences are routinely found when the input SNR is above a certain threshold (Supplementary Fig. 3). Taken together, the combination of optimized optical path, cofactors[39,51] and dedicated machine learning algorithms from the CARE family enabled the acquisition of high-SNR images at exposure times as low as 50 ms in living animals.

Inspired by these positive results, we set out to test the advantages of low-background autofluorescence recordings, and established transgenic *C. elegans* animals that express a green enhanced Nanolantern exclusively in the touch receptor neurons (TRNs). High-resolution imaging of these neurons is often precluded by the abundant autofluorescence that emanates from the ubiquitous gut granules under epifluorescent illumination (Fig. 4a). This is of particular concern in old animals[52], in which the signal of the autofluorescence can become more intense than the specific label, which makes it difficult to distinguish between

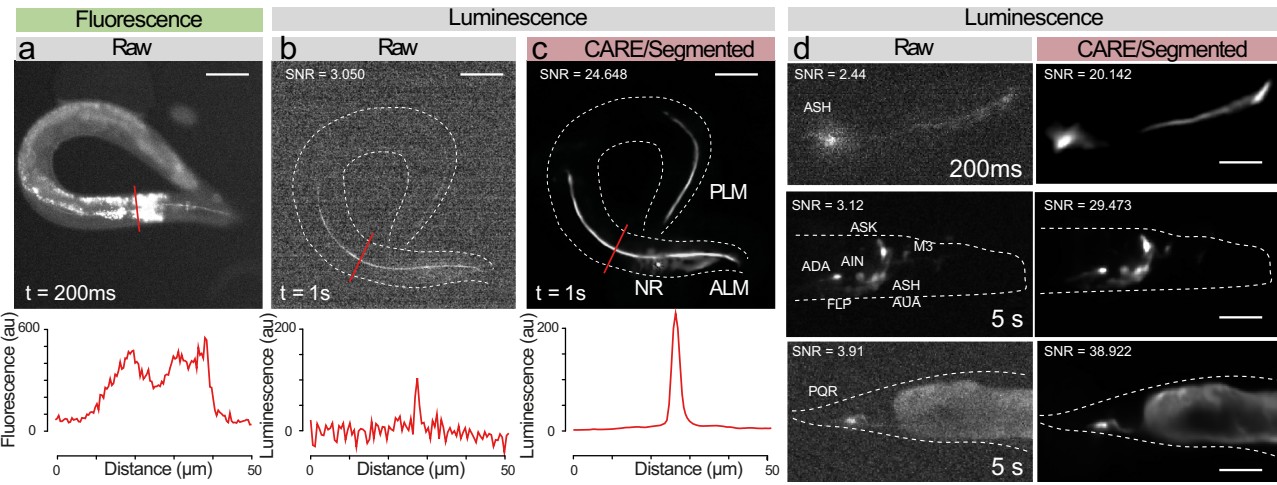

**Fig. 4 Content-aware restoration of photon-starved bioluminescent images. a–c** Suppression of autofluorescence in bioluminescent restoration microscopy. **a** Fluorescence picture of a worm expression mNeonGreen-enhanced Nanolantern (GeNL) in touch receptor neurons; Note the abundant autofluorescence originating from the internal tissues. **b** The same transgenic in bioluminescent contrast before and, **c** after CARE denoising. Scale bar = 50 μm. Lower panel shows intensity profile through the lines indicated in the upper micrographs. **d** Versatility of the neuronal reconstruction as shown on several neurons in *C. elegans*, such as ASH and a neuronal ensemble expressing the mTurquoise2-enhanced Nanolantern in glutamatergic neurons (*eat-4p*:TeNL) and PQR. Scale bar = 15–30 μm.

both. Indeed, the fluorescent images derived from animals expressing GFP in TRNs on a standard epifluorescence microscope showed extensive out of focus light due to background autofluorescence of the gut (Fig. 4a). In contrast, living animals that express the TRN::luciferase and were supplied with the optimized co-factor, a single, specific signal is visible from the monopolar neurites of these neurons, and no spurious autofluorescence could be seen (Fig. 4b). However, because of the small size of the TRNs, the obtained SNR is very low at exposure times as short as 1s. When we applied our previously trained CARE model to degraded bioluminescent images derived from transgenic TRNs in aged animals, our model was able to effectively enhance the SNR and cleanly visualize the neurons for further inspection (Fig. 4c, see Discussion). Despite the low SNR of the input image, the inference was astonishingly good. We also established that this approach is not limited to a specific neuron in *C. elegans*, and we have successfully enhanced the degraded bioluminescent images acquired for ASH, PQR, and vGLUT EAT-4 expressing neurons in the head (Fig. 4d). Strikingly, we found satisfactory performance of the model with exposure times as low as 200 ms taken on ASH, a neuron in the head of *C. elegans* with a diffraction-limited axon caliper in the range below <500 nm[53]. Taken together, we showed that we could achieve high-performance with a small, but diversified training dataset composed of 2500 different image pairs, that resulted in a generalizable and transferable model to infer noiseless images from severely degraded inputs of different cellular structures in *C. elegans*. Consequently, this allowed us to build our pipelines using free cloud-computing resources, which are accessible to a standard research laboratory[45].

**Combining CARE with various deep-learning methods**. To demonstrate that our combination of bioluminescent imaging and deep-learning can be generalized to other animals and biosystems, we generated bioluminescent zebrafish embryos expressing a membrane-bound red shifted Nanolantern and mounted them for imaging in our LowLiteScope at 4 h postfertilization (hpf) to collect a series of conventional z-stacks (see Methods). Even though the enveloping layer (EVL)—the outermost monolayer of cells surrounding the embryo in the blastula

stage—is clearly visible as a tessellated epithelial cell layer (Fig. 5a) in the confocal scanning fluorescence microscope pictures, a strong out-of-focus haze obscured the signal strength and SNR in bioluminescence contrast. Despite this challenging condition, we were still able to record a signal reminiscent of the cell junctions after 100 ms exposure time (Fig. 5b, Supplementary Fig. 4); however, post-processing such as segmentation proved difficult due to the poor SNR between the specific membrane label and unspecific background. Because this out-of-focus background is absent in the fluorescent images due to the optical sectioning of the confocal microscope, we reasoned that the blurred bioluminescent images can be reconstructed using a neuronal network trained on clean images derived from background-free confocal microscopy. Unfortunately, due to tissue movements during the acquisition time in the point scanning confocal microscope, we were not able to collect paired image stacks (degraded vs ground truth) to assemble a dedicated training pipeline from zebrafish embryos. We thus wondered if the bioluminescent zebrafish images can be restored with a pretrained CARE model, originally established to restore information from low-SNR images of epithelial monolayers in *Drosophila* wing discs[21], a tissue with similar tessellated morphology and signal distribution. We first tested that the monolayer morphology of the zebrafish EVL is similar to the wing disc. To formally approve the similarity, we trained a Siamese neuronal network with images of zebrafish epithelial cells (positive score) and random images (negative score) and calculated the Euclidean distance δ between them. A low δ means a high structural similarity between the two input images. Once sufficiently trained, we compared the unseen *Drosophila* wing disc image with the zebrafish dataset and found a very low δ indicative for a high similarity between the images (Supplementary Fig. 5). In contrast, images derived from *C. elegans* or images from a dataset with labeled nuclei yielded a high δ, indicative for a low structural similarity. We thus proceeded and restored the SNR of zebrafish images with the *Drosophila* model. Despite the challenging task due to the poor input SNR, we found that this model was generalizable and fit our input extremely well, being able to greatly improve the SNR (Fig. 5b, c) and remove the blur that originated from out-of-focus. Alternative and dedicated neural networks that learned to restore confocal data from widefield images[54], however, might even improve the

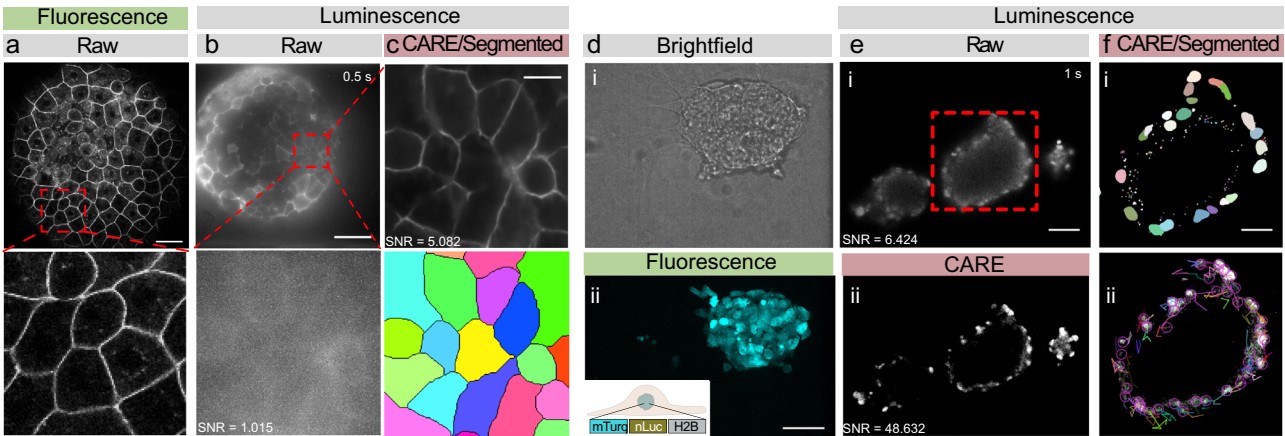

**Fig. 5 Seamless denoising and segmentation of bioluminescent zebrafish embryos and mouse embryonic stem cells. a** Laser scanning confocal fluorescence image of a 4 h post-fertilization (hpf) zebrafish embryo expressing membrane-bound GPI-GFP taken on a Leica SP5. The red box indicates the close-up below. Scale bar = 50 μm. **b** Unprocessed bioluminescence image of a 4 hpf zebrafish embryo expressing GPI-GFP targeted to the plasma membrane. The red square indicates the high magnification close-up below. Scale bar = 100 μm. **c** The same bioluminescent signal of the zebrafish embryo was restored using a pretrained CARE pipeline optimized for epithelial monolayers. The bottom picture shows the segmented bioluminescent image. No segmentation was possible on the raw image. Scale bar = 20 μm. **d** Brightfield (i) and fluorescence image (ii) of a spheroid of mouse embryonic stem cells. **e** Unprocessed raw (i) and denoised (ii) bioluminescence image of similar spheroids. Scale bar = 75 μm. **f** Segmented nuclei (i) after denoising, overlayed with their individual tracks throughout the timelapse (ii). Scale bar = 35 μm.

reconstruction efficiency in future. Critically, these signal restorations and improvements enabled the segmentation of individual cells in the embryo (Fig. 5c) which may afford the calculation of their perimeter and cell area and possibly increase the optical sectioning capability of the bioluminescence microscopy—a procedure that otherwise would not be possible.

We were next interested to demonstrate cellular dynamics in mouse embryonic stem cells and generated a stably transgenic cell line expressing a nuclear localized luciferase by fusing mTurquoise-NL to histone (see Methods, Fig. 5d). After optimization of the co-factor delivery (see Methods) we performed timelapse imaging of individual cells in spheroids from mESCs and recorded their nuclear dynamics (Fig. 5e, Supplementary Movie 2). To improve visual quality and the ability to quantify nuclear trajectories, we sequentially employed two published convolutional neural networks. We first passed the noisy images through a pretrained CARE neural network for denoising nuclear morphologies[21] and then performed nuclear segmentation with the StarDist algorithm[55] (Fig. 5e, f). This approach allowed us to track the migratory path for each individual nuclei within bioluminescent spheroids. Taken together, these approaches demonstrate the possibility to image subsecond dynamics of subcellular localized bioluminescent probes in *C. elegans*, zebrafish and mouse embryonic stem cells.

**Single-exposure volumetric calcium imaging in moving animals.** Up to this point, the long exposure times in bioluminescence imaging have largely hindered the acquisition of three-dimensional image stacks, especially in moving animals. Often, it is desirable or even important to obtain the whole 3D representation of a fast biological process, e.g., during calcium imaging of neuron or muscle contraction. We thus sought to establish single-exposure volumetric light field imaging[30] to quantify calcium dynamics in freely moving animals using bioluminescent calcium indicators. To do so, we equipped our LowLiteScope with a microlens array that is matched to the magnification and numerical aperture of the imaging lens and projected the entire light field onto the qCMOS sensor for plenoptic imaging in four dimensions (Fig. 6a)[56]. To obtain the volumetric information from a flat image, the light field needs to be deconvolved

computationally[56,57]. Traditionally, however, this process is computationally very demanding and takes up to several minutes for a single image[31,32] amounting to many hours or even days computing time for a whole time series, which makes the recording of cellular dynamics unattainable. Several AI-based algorithms have been proposed to speed up the deconvolution and enhance performance[24-26], that considerably outperform traditional light field processing (see Introduction). To create a neural network for the reconstruction of *C. elegans* expressing a fluorescent calcium reporter in the body wall muscles, we first trained a NN with synthetic light field data[24] as the input and experimental confocal stacks as the target (Fig. 6b). Both features were derived from the same immobilized animal and each stack was convolved with the light field point-spread-function (PSF) to generate the synthetic input data (Fig. 6b, and Methods). As described before[24], this approach shortened the processing time from 30 min to 100 ms per full frame image as compared to traditional lightfield deconvolution algorithms, but provided modest results with relatively high NRMSE and low SSIM indices (Fig. 6c, Supplementary Fig. 6).

We then extended the model using transfer learning with purely experimental data containing fluorescent light field images and *z*-stacks taken from the same sample as just described (Supplementary Fig. 7). This network knowledge expansion allowed us to be more specific to the experimental images we got from our setup, gave more flexibility to perform well for low and high-SNR light field images, reduced the possibility to obtain artifacts and improved the inference quality colorgreen (Fig. 6c, Supplementary Figs. 6, 7). It turned out that the inference quality was sensitive to the input SNR, as light field images that were acquired with short exposure times and yielded an SNR < 3.6, the light-field reconstructions were less reliable, as indicated by a low structural similarity (Supplementary Fig. 8). We found that exposure times of 5s in bioluminescence contrast are required to obtain a clear representation of the scene, but with blurred dynamics due to sample movement. In order to enable faster frame rates to 'freeze' animal movement and capture the full dynamics of the calcium dye, we applied the CARE pipeline for denoising the low-SNR lightfield images obtained at low exposure times prior to the light field deconvolution within a sequential

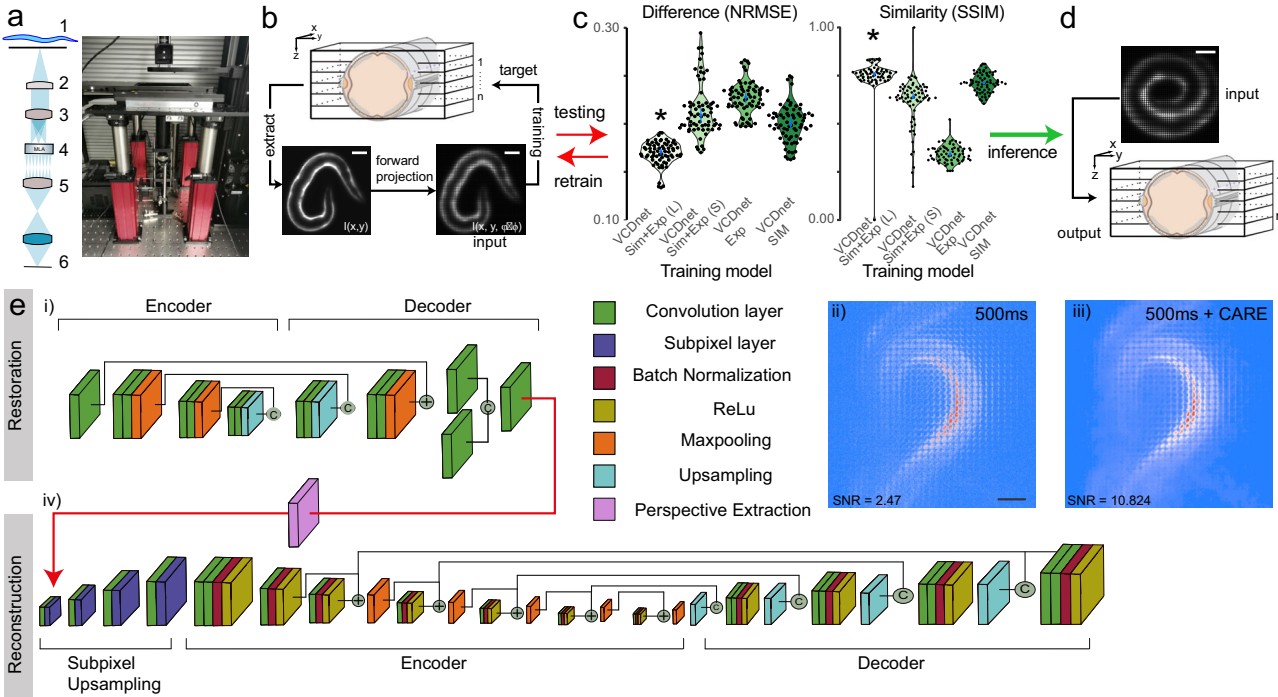

**Fig. 6 Single-exposure, volumetric bioluminescence microscopy. a** Schematic and photograph of the optimized Low-LightField microscope. (1) Sample, (2) Objective, (3) Tube lens, (4) Microlens array, (5) Relay lens, (6) camera. **b–d** Training pipeline to obtain fast deconvolution of 2D experimental lightfield data into 3D image stacks. **b** A 3D image stack was acquired on fluorescent samples representative of the bioluminescent signal in the final experiment. The stack was convolved with the lightfield PSF, to obtain a synthetic lightfield image, which was subsequently used to map onto the 3D ground-truth stack (see also Supplementary Fig. 7). The training quality of the individual models (**c**) was tested against unseen samples by calculating the difference and similarity to the ground truth ($N = 70$ images/model). The best model (indicated with an asterisk, $p < 1.2\mathrm{e}{-}9$ compared to the 2nd best model, Wilcoxon test) with the lowest error and highest similarity was used to **d** reconstruct experimental bioluminescence images. Scale bars = 50 μm. Blue dots in **c** indicated median ± 95% confidence interval. **e** Pipeline for bioluminescence reconstruction. An initial (i) CARE denoising step is used to increase the SNR of (ii) noisy bioluminescent lightfield images. The individual layers are color coded according to their function. The clean images (iii) are fed into the VCD network[24] (iv) after perspective extraction to reconstruct the 3D information. Scale bar = 50 μm.

application of two neural networks (Fig. 6e). To do so, we connected the two networks after perspective extraction (Supplementary Fig. 9). The denoising led to a striking increase in SNR of the light field image, and with this approach, we were able to obtain substantially better reconstructions than without (Supplementary Fig. 10).

Using these improvements in training and network concatenation, we were able to obtain three-dimensional calcium recordings from whole animals with typically 200–500 ms exposure time per light field image, which we reconstructed to create a full 3D stack of the bioluminescent scene represented by 31 z-planes with a spacing of 1.5 μm between sequential planes (Fig. 7a, Supplementary Fig. 10). Critically, at an imaging speed of 5 volumes/second, this is equivalent to 6.4 ms exposure per frame if it were acquired with traditional volumetric imaging (e.g., sequential planes in a confocal z-stacks). In these bioluminescent calcium recordings, we observed higher intensity on the concave side of the bend, consistent with high-calcium concentration during muscle contraction (Supplementary Movie 3). Importantly, the reconstructions preserved the relative intensity distribution within the sample, as we did not find noticeable differences between the reconstructed forward projection and the ground truth (Supplementary Fig. 6). We also observed that most calcium signal comes from equatorial region of the muscles and rapidly drops off towards the lateral sides (Fig. 7b). This implies that the contractile power is localized to the equatorial regions, where the largest bending moment can be applied. Consistent with a high-calcium concentration during muscle contraction, we observed

that the largest intensities mapped to positive body curvatures (Fig. 7c).

Taken together, we have demonstrated the performance of an improved bioluminescence microscope on living tissue culture cells for subcellular labeling of actin and microtubules, zebrafish epithelial cell organization and nuclear dynamics in mouse embryonic stem cells. Lastly, we combined a sequential neuronal network composed of content-aware image restoration pipelines and light field reconstruction to enable high-speed, subsecond volumetric imaging of a genetically encoded calcium sensor in freely moving animals.

## Discussion
Here, we have shown that a combination of an optimized optical path and advanced computational tools substantially improves the SNR of bioluminescence microscopy that rivals that of conventional fluorescence microscopy to resolve fast biological dynamics in three dimensions.

Despite the promise of deep-learning-based methods to complement, assist and enhance bioluminescence microscopy, several challenges and pitfalls are frequently encountered. Apart of the common issues related to deep learning, which are outlined in ref. [58], and concern training sample size, hyperparameter choice, or network architecture, some specific points need to be observed in excitation-free bioluminescence microscopy.

(1) The constitutive light emission of the luciferases in presence of the co-factor lead to an extended source of light, which

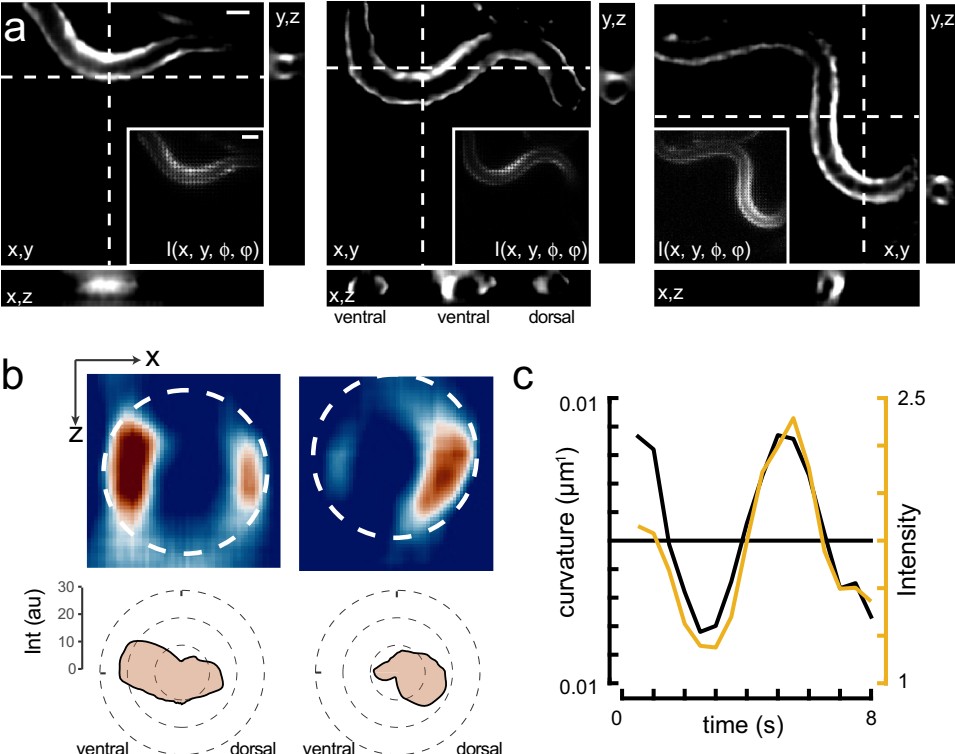

**Fig. 7 3D calcium imaging of freely moving animals. a** Sequence of reconstructed 3D images of a moving animal showing high-calcium activity at its contracted side. Images show a single plane of the reconstructed z-stack. Inset corresponds to the raw lightfield image. Scale bars = 40 μm. **b** Sideview image of the curvature-dependent calcium signal in muscles during ventral and dorsal body bends with warmer colors representing higher calcium signals. Scale bar = 20 μm. The polar plot shows the intensity distribution on the ventral and dorsal side. Dotted line corresponds to the circumference of the animal. **c** Intensity of the bioluminescent calcium indicator and curvature variation on the ventral side during animal crawling under the lightfield microscope. Black = curvature, yellow = calcium signal.

limits optical sectioning of thick samples. The consequence is a higher out-of-focus background blur in samples that are much thicker than the point-spread function of the detection objective. Further, many modalities in fluorescence microscopy rely on point-spread function engineering (STED) of the excitation light beam or modulation of the excitation intensity (SIM) and incident angles (TIRF, light sheet microscopy). Due to the absence of an excitation light beam in bioluminescence microscopy, fundamentally different approaches need to be taken to break the diffraction barrier[59] or create optical sectioning[54]. The combination of bioluminescence microscopy and neural networks trained on fluorescent samples, as presented in this study, might provide an entry point for further improvement in spatiotemporal resolution.

(2) As stated in the introduction, the bioavailability and distribution of the co-factor within animal tissues is a limiting factor. In our experiments we noted these artefacts in the stem cell spheroids, where we observed a predominantly cortical signal originating solely from the superficial cells in the aggregate. Such effects will be more pronounced in brain tissue of small animal models where the blood-brain barrier hinders transport into neuronal cells[60]. Likewise, the *C. elegans* skin is a permeability barrier, and mutations that lead to less crosslinking without affecting animal health, e.g., *bus-17*[61], might facilitate the bioavailability of the co-factor inside animal tissues.

(3) Despite the impressive performance of the CARE pipelines on the presented test data, the input images require a certain information density. If the input SNR of the test images is too low, a restoration can be expected that might

not sufficiently satisfied pre-established criteria (Supplementary Figs. 3 and 8). In that case, a higher exposure time should be employed.

(4) The generalizability of the models needs to be determined a priori. At best, for each biological structure, a dedicated model is available. However, unwanted autofluorescence often precluded the acquisition of a high-quality, artefact-free training dataset, especially if fluorescence emission from the specific label is weak and spatially overlaps with autofluorescence. This complicated the construction of a dedicated model for neuronal morphologies (see also discussion below).

(5) 4D imaging of cellular dynamics. Due to the longer exposure times, traditional 3D imaging of bioluminescent samples using z-stacks is often slower than the underlying biological dynamics, which leads to blurred out detailed and motion artefacts. Thus, single exposure, bioluminescence light field microscopy substantially increases the time resolution and captures calcium dynamics in freely moving animal.

Because the whole sample usually emits light, light field microscopy is the perfect partner in crime with bioluminescence to obtain a 3D representation of the sample with a single exposure. Traditionally, the resolution of light field images is bound to the point-spread function of the lenslet array, which, after deconvolution gives access to micron sized image features. Deep-learning models to reconstruct light field images, however, have obtained far better resolution than what is expected from the design of the lenslet array[24]. This is due to the training process, which find a complex transfer function between a low-resolution

light field image and a high-resolution ground-truth image stack (Supplementary Fig. 11). Importantly, latter has the nominal spatial resolution that is determined by the diffraction optics of a standard optical microscope and can be up to 10 times higher than the lenslet array employed in a light field microscope[56]. Therefore, the neural network learns the how the pattern in the lightfield images gives rise to a super-resolved 3D representation (Supplementary Fig. 11b). Likewise, the deep-learning process also is able to improve the angular resolution and angular field, depending on the design and quality of the training data. With these improvements, the deep-learning reconstruction approaches the ground-truth images (after rectification). Still, since the rectification involves a downsampling, it decreases the spatial resolution compared to the raw fluorescence image stack (Supplementary Fig. 11), leading to an overall degradation of the imaging performance.

Supervised deep-learning methods invariable outperform self-supervised methods—also for bioluminescence microscopy restoration. Thus, unless a generalizable model is available, the restoration of each specific structure requires different models—a feat that is not always within reach. For example, due to the abundant autofluorescence in *C. elegans*, we have not been able to generate a suitable, dedicated training dataset composed of paired noisy and ground-truth data of neurons (due to their low fluorophore expression compared to muscles). Often though, properly trained models are transferrable as long as the underlying sample structure and intensity distribution is similar. How do I know if the pretrained models accept my input data? To answer and guide the choice, we have thus set up a classification procedure based on Siamese neural networks (Supplementary Fig. 5) to formally distinguish between images of a different classes[62]. For example, StarDist algorithms for nuclear segmentation work perfectly independent of whether the sample comes from mouse stem cell or *C. elegans* nuclei. Likewise, given a similar morphology between zebrafish EVL and *Drosophila* epithelia, existing models might be transferable between these structures even though they are from a different organism. We applied the same reasoning and used a model trained on a dataset for muscular morphologies to neurons. The comparison of representative input images yielded a low Euclidean distance between the intensity distribution (Supplementary Fig. 5), predicting a similar performance of the CARE algorithms. Indeed, the restoration proved to work well in the majority, although we noted that the denoising of the images depicting the neuronal morphologies appeared to broaden the width of the axonal caliper. We attributed this effect to the fact that we used a machine learning model trained on muscle images that often contain wider intensity distributions. Unfortunately, we were not able to establish a dedicated training pipeline for neuronal morphologies. Due to the low signal obtained from the transgenic animals with the neuronal fluorescent labels, the autofluorescence arising from other tissues had a dramatic impact in the ground-truth images—the ground truth was contaminated with unwanted background autofluorescence. Thus, the neuronal network learned to recognize these autofluorescent signal even in bioluminescent images, where they were absent. In future, dedicated models[63] will be needed to assume perfect performance, that involve an automated discriminator or 'forgetting' layers.

To unlock the full potential of bioluminescence microscopy, novel luciferases and cofactors will be needed to obtain single-cell resolution at high magnification in crowded tissues, e.g., organoids or for whole brain luminescent calcium imaging. In the future, spatiotemporal resolution and light-capturing ability could further be improved through combinations of wavefront coding[64], tunable optics[65], Fourier-lightfield microscopy[66], and new transformer networks that are trained to provide a spatially

super-resolved representation of the scene[67]. Our results pave new avenues for excitation-free, noninvasive low-light imaging in microscopy, diagnostics, and biomedicine.

## Methods

**C. elegans culture**. Animals were maintained on Nematode Growth Medium (NGM) plates seeded with *Escherichia coli* OP50 bacteria using standard protocols[68]. Where indicated, age-synchronized animals were used and handled as described[69]. The following strains have been used: MSB266[*eat-4(ky5)III; hpIs166; mirEx89[pNMSB26(sra-6p::sng-1::TeNL::unc-54 3'UTR)]*], MSB343[*mirEx123(myo-3p::TeNL::unc-54)*], MSB426[*mirSi16 II; eat-4(mir28)III; lite-1(ce314)X, mirEx168[pNMSB17(eat-4p::TeNL)]*], MSB557[*mirEx218[pNMSB52([mec-4p::GeNL::let-858 3'UTR)]*], MSB577[*bus-17(br2)X; mirEx123[pNMSB40(myo-3p::TeNL250::unc-54)]*], MSB1041[*bus-17(br2)X; mirIs92[pNMSB55(daf-16p::daf-16::GeNL::unc-54)]*]

**Molecular biology**. All plasmids used for this study were constructed using the Gibson assembly method. pNMSB17, pNMSB26, and pNMSB40 plasmids carrying the mTurquoise2-enhanced Nanolantern (TeNL) have been described elsewhere[34]. To generate pNMSB52 (TRN:GeNL) a mNeonGreen-enhanced NL (GeNL) gene was cloned with primers 5'-atggtctccaagggagaggaggacaac-3' and 5'-TTACGCGAG GATACGCTCGCAGAGAC-3' into a plasmid containing the *mec-4* promoter. pNMSB55 (daf-16GeNL) was constructed using primers 5'-gccaagctatgaat tcaacttgagcatctctttttcttgg-3' and 5'-gttgtcctcctctcccttggagaccatcaaatcaaaatgaatatg ctgccctcc-3' to amplify daf-16 promoter and ORF and clone then into a plasmid containing GeNL.

**Transgenic mouse embryonic stem cell line generation and maintenance**. To generate the transgenic mouse embryonic stem cell (mESC) line, $1 \times 10^6$ G4 cells[70] were lipofected using Lipofectamine™ 3000 Transfection Reagent (Thermo Fisher Scientific, L3000001) mixed with 0.625 µg of PX330 (Addgene, 98750) (sgRNA 5'-ACTGGAGTTGCAGATCACGA-3') and 1.875 µg of circular HDR template. Lipofection was carried out using a PenStrep-free mESC medium (described below). Cells were single-cell FACS-sorted for mTurquoise expression using a BD Influx Cell Sorter (646500KZ, model Influx V7 Sorter, software BD FACS Sortware). Individual clones were screened by PCR amplification.

mESCs were maintained and expanded on 0.2% gelatin-coated dishes in mESC media composed of the following: Knock-Out DMEM (Thermo Fisher Scientific, 10829-018) supplemented with 15%Fetal Bovine Serum (FBS) (in-house mESC-tested), 1000 U/ml Leukemia Inhibitory Factor (LIF, in-house generated), 1 mM Sodium Pyruvate (Thermo Fisher Scientific, 11360070), 1× MEM Non-Essential Amino Acids Solution (Thermo Fisher Scientific, 11140050), 50 U/ml penicillin/ streptomycin (Thermo Fisher Scientific, 15140-122) and 0.1 mM 2-mercaptoethanol (Thermo Fisher Scientific, 31350010). Cells were cultured at 37 °C with 5% $CO_2$. Medium was changed every day and cells were passaged using 0.05% Trypsin-EDTA (Thermo Fisher Scientific, 25300054) and quenched 1:5 in DMEM supplemented with 10%FBS (Life Technologies, 10270106). Cells were tested monthly for mycoplasma contamination by PCR.

**Tissue culture experiments**. The luminescent markers clathrin:CeNL, actin:YeNL, and the plasma membrane marker lyn::OeNL[14] were transfected into HeLa cells with Lipofectamine using standard procedure. Three days post transfection, tissue culture cells were supplemented with Hikarazine co-factor (at 0.8 µg/ ml) and imaged in the LowLiteScope. CeNL-Clathrin_pEGFP, ReNL-Actin_pcDNA3, and Lyn-OeNL_pcDNA3 was a gift from Takeharu Nagai (Addgene plasmid # 89540, # 89531, # 89528).

**Zebrafish**. Zebrafish (*Danio rerio*) were maintained as previously described[71]. Embryos were kept in E3 medium between 25 and 31 °C before experiments and staged according to morphological criteria[72] and hours post-fertilization (hpf). All protocols used have been approved by the Institutional Animal Care and Use Ethic Committee (PRBB-IACUEC) and implemented according to national and European regulations. All experiments were carried out in accordance with the principles of the 3Rs (replacement, reduction, and refinement).

AB wild-type zebrafish embryos were micro-injected at 1-cell stage with 100 pg GPI-GFP mRNA. mRNA was synthesized using the mMessage mMachine Kit SP6 Kit (Ambion AM1340M). Embryos were dechorionated at sphere stage (4 hfp) and mounted in 1% low-melting point agarose in Danieaus solution (58 mM NaCl, 0.7 mM KCl, 0.4 mM MgSO4, 0.6 mM Ca(NO3)2, 5 mM HEPES) on 35 mm glass bottom dish (MatTek). Embryos were imaged with a ×20 glycerol-immersion objective on Leica SP8 confocal microscope. Laser excitation of 488 nm and HyD detector were used. Z-stacks of 0.2 µm spacing between z-slices were acquired.

**Bioluminescence microscopy: instrument design**. We redesigned a standard widefield microscope and with a minimal set of components to minimize photon loss. We used a ×40/1.25 silicon immersion lens (Olympus) as imaging objective, and a 100 mm tube lens (ASI Imaging) to project the collected light onto a single

photon-resolving camera (Hamamatsu Photonics Orca Quest, or Fusion). The sample was mounted on a standard K-frame on top of a motorized Maerzhaeuser stage and a tissue culture incubator for temperature and $CO_2$ control. To keep the sample in focus, an objective piezo-positioner was used (MIPOS 500, Jena Piezo-systems), in conjunction with an autofocus system (CRISP, ASI Imaging). The whole setup was controlled with PycroManager[73], calling µManager out of Python.

For single-exposure volumetric imaging we added to the setup a microlens array ($200 \times 200$ lenses) with a pitch of 222 µm and focal length, a radius of curvature of 0.85 mm, a focal distance of 1.86 mm and a size of $11 \times 11 \times 1.5$ mm (Okotech APO-Q-P222-F1,86 (633)). We chose these specific properties to match the NA of the described objective to ensure optimal spacing between the lenslets. To align the focal plane of the microlens array with the image plane of the telescope, we used a relay lens composed of two lenses of 100 mm focal length. We used a zoom housing system to modify the distance between the CMOS sensor and the lenses in order to calibrate the setup before the image acquisition.

**Bioluminescence microscopy: image acquisition**
*Mouse embryonic stem cells (mESCs).* Days prior to imaging, $10^6$ transgenic mESCs were plated on a 35 mm No. 1 glass bottom µ-Dish (Cellvis, D35-10-1-N) coated with 0.1% gelatin (Stem Cell Technologies, 07903). Cells were 40–85% confluent during imaging, providing variation in three-dimensional colony (spheroid) volume and size. Cells were washed with PBS and new mESC media was added immediately prior imaging to eliminate debris. To image bioluminescence, the LowLiteScope was equipped with environmental control to maintain 37 °C and 5% $CO_2$ within a top stage incubator (LCI incubator system, Gas Mixer CA-10 and temperature controller TP10) and a qCMOS Orca quest C15550-20UP camera. 2–4 µM fluorofurimazine (FFz, Promega) subtrate was perfused through the cell culture dish, using a peristaltic pump, at a rate of 1 mL/min. Bioluminescence was observed immediately following substrate perfusion. For timelapse imaging, fresh substrate was added every 45–60 min. Z-stacks of 5 µm spacing were acquired every 5 min with an exposure time of 500 ms per slice.

*Zebrafish.* The membrane-bound orange enhanced Nanolantern was subcloned from Lyn-OeNL_pcDNA3 (Addgene plasmid # 89528) into a GPI-anchor containing plasmid. To generate bioluminescent Zebrafish embryos, 100 pg of lyn::OeNL mRNA was injected in 1-cell stage embryos to visualize the plasma membrane. Approximately two hours post-fertilization (2 hpf) at the 64-cell stage, embryos were dechorionated manually using a pair of forceps[74] and incubated in 400 µM fluorofurimazine (FFz, Promega) for 2 h. At sphere embryonic stage (4 hpf), embryos were mounted in a 35 mm No. 1 glass bottom µ-Dish (Cellvis) using 1% low melting point agarose (Promega) in Danieauś solution (58 mM NaCl, 0.7 mM KCl, 0.4 mM $MgSO_4$, 0.6 mM $Ca(NO_3)_2$, and 5 mM HEPES). Before agarose polymerization, embryos were oriented with blastomere cells facing to the glass. Bioluminescence imaging was performed with a ×40/1.25-NA silicon oil-immersion objective, a 100 mm tube lens (ASI Imaging) and a Hamamatsu Orca-Fusion camera (C14440-20UP) was used. Z-stack images of the embryo were taken with 2 s or 0.5 s exposure time every 5 µm, resulting in a depth approximately of 155 µm.

*C. elegans.* Bioluminescence images were acquired on the optimized LowLiteScope described above. If otherwise stated, L4 or young adult animals were mounted in a 1% agar pad and treated with citrate buffer (pH 6.5), 20% DMSO, 0.05% Triton X-100, and 1.25% pluronic F128 and the chemical co-factor Hikarazine[51] for the generation of a bioluminescence signal.

*Daf-16 in C. elegans.* Bioluminescence DAF-16 dynamics were acquired on the optimized LowLiteScope described above. Young adult animals were mounted in a 1% agar pad and 1 µL of 20 µM fluorofurimazine (FFz, Promega) was added directly prior to imaging without preincubation time. We acquired the dynamics with and without exposing the animal to a heat shock. We used a stage-top incubator system T (LCI), to maintain a temperature of 37 °C during imaging.

Fluorescence DAF-16 dynamics were acquired on a Leica DMi8 inverted fluorescence microscope with a Hamamatsu Orca Flash4.0 V3 sCMOS and a ×40/1.1 water-immersion lens (Leica). Animals were mounted in a 1% agar pad. Same as above, we acquired the dynamics with and without exposing the animal to a heat shock. We used an incubator (Warner Instruments) to maintain a temperature of 37 °C during imaging. Animals were excited at 488 nm using a Lumencor SpectraX LED illuminator guided through a triple band pass dichroic (FF459/526/596-Di01-25x36, Semrock Co.) and images were taken every 30 s with a 100 ms exposure time.

For final quantification, the intensity of an ROI drawn around the nucleus was divided by the the intensity of the cytoplasm close by, except for the unstressed bioluminescence animals where the nucleus was not visible. The nuclear/cytoplasmic ratio was plotted in R and displayed in Fig. 2c.

**Machine learning: content-aware restoration (CARE).** CARE has proved that is possible to effectively denoise fluorescence microscopy images under conditions of low light or low exposure time[21]. One source of the distortions in low-SNR fluorescence images is typically limitations of the camera readout noise, photon noise or the resolution loss due to under-sampling. Image denoising is the process of separating the signal $s$ and the signal-degrading noise $v$ of a distorted image $x$. The noisy image can be thought of being the result of a function $x = f(y)$ applied to the ground-truth (GT) image $y$. The inverse function, e.g., $y = f^{-1}(x)$ is typically computationally demanding or intractable to describe mathematically. Instead of calculating the real denoise function $f^{-1}$, CARE learns to map an approximation of this function by processing a large set of pairs $(x, y)$ of noisy images $x$ and their corresponding true images $y$ by using a CNN based on residual U-net architectures[21].

**Denoising bioluminescent body wall muscle and neurons in *C. elegans*.** For training data acquisition, L4 transgenic *C. elegans* expressing mTurquoise fused to nanolantern (MSB343) were imaged on a Leica DMi8 inverted fluorescence microscope with a Hamamatsu Orca Flash4.0 V3 sCMOS using different camera exposure times. For imaging, an epifluorescence illumination with a ×10/0.3-NA or ×25/0.95-NA (numerical aperture) water-immersion objective was used. Fluorescent animals were excited at 430-nm using a Lumencor SpectraX LED illuminator guided through a triple band pass dichroic (FF459/526/596-Di01-25x36, Semrock Co.). To obtain a low-SNR image representing the bioluminescence setup, we used a low exposure time (4 ms), and to obtain a high-SNR image representing the 'ground truth' and the desired inference quality of our model, we used a high exposure time (100 ms). Both channels were acquired simultaneously through a Hamamatsu Gemini W-View optical beamsplitter equipped with a CFP/Venus emission filter set for the ground truth and degraded channel, respectively. The two images were cropped and superpositioned in python as part of the preprocessing pipeline prior to the training. Importantly, both channels were recorded at the same time allowing us to use freely moving animals as training sample. To achieve a high variety of postures, animals were mounted in a "pool" created with 1% agarose pad and halocarbon oil 700 (Sigma–Aldrich) to capture different postures of the worm, recapitulating crawling movement. This method enables dataset diversification, permitting the model to generalize in freely moving animals. We collected 2500 pairs of images of average size $1024 \times 512$, resulting in a mere ≈ 9 GB of training data.

From the 2500 images, we extracted subimages (patches) which were given to a U-Net type architecture (Denoising CARE 2D topology) for training and validation[21]. We tried different hyperparameters to optimize the configuration of the model and consequently, improve the model performance, e.g., minimize the loss function. After training we evaluated the models with 100 unseen images of *C. elegans* at various positions. Please refer to Supplementary Fig. 1a for more details. To validate the quality of the prediction, we compute the NRMSE and SSIM against unseen GT images.

**Denoising, segmentation, and tracking of mouse embryonic stem cell.** To restore the bioluminescence stem cell timelapse, we use the *human U2OS cells* dataset stained with Hoechst 33342 markers and imaged with a camera exposure range of 15–1000 ms provided by the authors of ref. [75]. We followed[21] training protocol and inferred our embryonic stem cell data with the best model checkpoint we obtained. We saw a considerable improvement after the processing of every frame in our timelapse. Exploiting this, we further analyze processed the data by using a pretrained Stardist segmentation algorithm[55] and TrackMate tracking algorithm[76].

**Denoising, projection, and segmentation of bioluminescent zebrafish embryo.** All zebrafish images presented in the article were acquired on transiently transgenic embryos, ≈4 hpf, expressing membrane-bound Nanolantern as described above. To restore the raw zebrafish bioluminescence images, we leveraged an existing training dataset with a similar morphological characteristic as the zebrafish epithelia. We cross verified the similarity of the input images and the training images using a Siamese neuronal network (see below). Specifically, we use the *D. melanogaster* with the membrane marker Ecad::GFP data provided by the authors of ref. [21]. We followed their training protocol and inferred our zebrafish luminescence data with the best model checkpoint we obtained. After the projection using the neural networks, we recovered most of the lost cellular membrane information allowing us to perform further analysis such as cellular segmentation (Fig. 5). To do this, we applied a morphological dilation filter with kernel size of 3 to join the small gaps where the membrane is not fully complete. Next, we perform watershed segmentation using the Fiji plugin MorphoLibJ to get single-cell segment distribution[77].

**Denoising light field images.** To denoise light field images we collected a bioluminescent dataset using anesthetized transgenic *C. elegans* under 1 µl of levamisole 2.5 mM and treated with 1 µl, 20 µM of fluorofurimazine (FFz, Promega) mounted in 1% of agar pads expressing mTurquoise fused to nanolantern at different developmental stages e.g., L3, L4, and young adults. All subsequent data acquisition was done with the same animal and protocol. To diversify the input distribution of the model, we acquired images with different camera exposure times e.g., 200 ms, 500 ms, 1 sec, 5 sec, and 10 sec sequentially, using the same field of view and pairing each image condition to the image with the highest exposure time we collected. For each position and condition we acquired a stack to collect the information of how the light changes through the lenslets over different z positions.

For the 3D stacks we took 3 planes above and below the current position with a step size of 1 $\mu$, thereby disentangling the inference model quality from the z position of the bioluminescent sample. For imaging, a bioluminescent light field setup described above was used. We collected 2034 paired of images with a size of $934 \times 976$, resulting in ~7 GB. We employed the photon number resolving mode to decrease the readout noise for both, training and testing datasets. Next, we employed linear transformations to regularize and prevent overfitting. We used the CARE 2D denoising training topology, extracting 508500 random subimages with a size of $128 \times 128$ and a batch size of 64. The training took 15 h and 32 min in a workstation equipped with Intel(R) Xeon(R) Gold 6248R CPU, 128 Gb of RAM and Nvidia Quadro RTX8000 48GB. All subsequent light field denoising and reconstruction training were done on the same setup. We evaluated the denoised images by calculating the SSIM and NRMSE metrics in unseen long exposure bioluminescent images (Supplementary Fig 5).

**Machine learning: bioluminescence lightfield reconstruction**. For bioluminescence light field reconstruction we use the neural network VCD-Net[24]. VCD-Net is based on a U-Net topology composed of an encoder-decoder sampling. The views extracted from the light field images (see Fig. 3) are transformed from perspectives to channels intrinsically done by the computation of the convolutional layers. Same as in the original U-Net architecture skip connections are defined between the encoder and decoder to preserve unprocessed information for better fitting. The network gradually transform the extracted views from the original light field image into the conventional 3D stack. The number of filters or dimensionality of the output space in the last convolutional layer is set to match the desired amount of planes for the 3D reconstruction.

**3D reconstruction with synthetic light field data**. For the data acquisition for the reconstruction of high-quality 3D data from synthetic light field images, we collected high-quality 3D stacks, an epifluorescence microscope with ×40 × /1.1-NA (numerical aperture) water-immersion objective and 488 nm excitation wavelength were used. We captured 26 different 3D stacks (of size $1024 \times 1024 \times 31$). Before the generation of synthetic data, we performed linear transformations, e.g., flipping, rotating, and inverting the z-axis to augment the dataset. Subsequently, we compute the simulated PSF through the microlens-array using ref. [32] to perform a light field projection to the acquired 3D stacks using ref. [24]. This process generates the synthetic light field images that correspond to the input in our neural network. Like the CARE pipeline, the training was done by gradually varying the model coefficients in order to minimize the loss function. For training we extracted 7518 pairs patches of size $176 \times 176 \times 31$ pixels and $176 \times 176$ pixels for the 3D target and input respectively. We trained for 100 epochs which represented a cost of 7.5 h in a single graphical processing unit. To evaluate the model reconstruction and demonstrate that the relative intensities are preserved, we calculated the SSIM and NRMSE maps against unseen fluorescence stacks. Furthermore, we compare the intensity profile on one edge of the worm to compare the intensity behavior between the light field reconstruction and the ground truth (see Supplementary Fig. 4).

**3D reconstruction with experimental light field data via transfer learning**. One of the main weaknesses about training purely with synthetic data is the learning limitation of experimental information that your setup might suffer, such as noise or lens misalignment. Therefore, minor errors introduced in the input might generate artifacts in the prediction. We thus set up a transfer-learning pipeline to generate a more realistic representation of the experimentally derived bioluminescence dataset. The overall procedure is displayed in Supplementary Fig. 7 and described as follows. Because light field images and widefield stacks represent optical information in different ways, performing registration between a couple of images in the dataset usually fails. Therefore, the reconstruction of the light field images needs to be made before proceeding to the registration[26]. As we previously stated, the 3D reconstruction of light field images is extremely time consuming. Hence, creating a dataset that consists solely on experimental data where it has enough information to cover the 3D LF reconstruction manifold would represent a gigantic effort. Therefore, we made use of the neural network trained purely with simulated data. For the imaging we used the commercial epifluorescence microscope Leica DMI8 with two available camera ports. In one port, we acquired light field images where we mounted a microlens array of pitch 192 and an effective focal length of 3.17 mm (APO-Q-P192-F3.17 (633)) in an sCMOS board level camera C11440-62U (Hamamatsu Photonics). The second camera port was dedicated to acquire the corresponding stacks with a CMOS ORCA-Fusion C14440-20UP. An objective ×40/1.1 water-immersion lens (Leica) and a 488 nm excitation were employed. We used four different exposure time imaging conditions, e.g., 1, 10, 50, and 100 ms for both, stacks and light field images. The same field of view was imaged using light field and epifluorescence sequentially. All conditions were paired to the highest exposure time as GT. A rectification on the ground-truth image stack was performed to ensure that the pixels in the light field image and the pixels in the ground-truth stack match perfectly and contain the same information. In other words, the 3D information for a given emitter in the GT was thus matched to the positional information in the light field image. This rectification lead to a resampling of the ground-truth images and the input light

field images that was accompanied with a loss of resolution (Supplementary Fig. 11).

We first predicted the experimental light field images to get a rough reconstruction of the structure and then apply an image registration with their corresponding experimental stack. To do so, we used a scaled rotation, which performs a translation, rotation and scaling of the images to find the correct transformation matrix and optimize the alignment, implemented in pystackreg[78]. By applying the same transformation to the experiment light field image, the matching training pairs were obtained. In that way, we were able to generate an experimental training dataset composed of ~11 GB in matter of hours. We then loaded the weights of the pretrained network with synthetic images and trained again with the well aligned images using the same hyperparameters as we discussed in the previous section. To evaluate the training, we once again calculated the SSIM and NRMSE metrics against a fluorescent target as our GT. We also compared the reconstruction intensity profile against the GT (Supplementary Fig. 4).

**Bioluminescence light field inference**. The bioluminescence light field inferences were made by rectifying the light field image (see below) and applying both pipelines consecutively. In other words, aiming to obtain the 3D reconstruction of the bioluminescence scene, the bioluminescence light field image needs to be denoised, rectified and reconstructed respectively. First, the trained 2D denoising CARE pipeline was applied on the raw bioluminescence image to improve the quality and recover lost information due to the noise. Then, we used the open source program LFDisplay[30] to calibrate and locate the microlens array. Next, the rectification process was done in the software provided by ref. [32] consisting in a resampling operation to contain the $11 \times 11$ angular light field views per lenslet (this number of angular points proved to work well and are used also in refs. [24,32]). It is used to make sure the perspective extraction is taking the pixel corresponding for a specific lenslet. If this is not done, the perspective extraction would not work, and would take shifted/blurred perspectives. Finally, the VCD-Net trained with experimental data was used to generate the 3D bioluminescent scene.

**Machine learning: normalization and quantification errors**
*Image normalization*. Before training and predicting an image, it is important to normalize it to have a comparable range of intensities. We used the widely used percentile-base normalization[21]

$$N(u; p_{\text{low}}, p_{\text{high}}) = \frac{u - perc(u, p_{\text{low}})}{perc(u, p_{\text{high}}) - perc(u, p_{\text{low}})}, \tag{1}$$

where $perc(u, p)$ is the p-th percentile of all pixel values of $u$. This normalization is done before feeding the images to the network either for training or inference.

*Image quality evaluation*. A common image quality metric to compare two images it is mean squared error (MSE) which measures the average of the squares of the errors;

$$\text{MSE} = \frac{1}{n} \sum_{i=1}^{n} (y_i - \tilde{y}_i)^2, \tag{2}$$

however, this metric cannot be used without normalize the images since the predicted image $\tilde{y}$, differ considerably compared to the ground-truth image $y$. To overcome this, we use the normalized mean squared error NRMSE, (range 0–1, lower is better), an image quality assessment defined by ref. [21]. For this, we applied equation (1) and parameterized the prediction with $\gamma(\tilde{y}) = \alpha\tilde{y} + \beta$ to scale the restored image and find the MSE metric where is minimal.

$$\text{NRMSE}(y, \tilde{y}) = \sqrt{(\text{MSE}(\gamma(\tilde{y}), N(y, 0.1, 99.9)))} \tag{3}$$

where

$$\alpha, \beta = \underset{\alpha', \beta'}{\arg\min} \text{MSE}(N(y, 0.1, 99.9), \alpha'\tilde{y} + \beta'). \tag{4}$$

To compare structural similarity, we use SSIM[79], which is a measure of the similarity between two images. It has a value of 1 if the images are identical and 0 if they are completely different. We applied the same normalization procedure for the SSIM calculation.

**Machine learning: Siamese network training**. A Siamese Neural Network (SNN) is a class of neural network architectures that contain two or more identical sub-networks. They have the same configuration with the same parameters and weights and it is used to find the similarity of the input images by computing the difference in their local intensity distributions.

While deep neural networks need a large volume of data to train on to classify input images into a binary yes/no category, SNNs learn a similarity function. This means that we can use an SNN to evaluate if two images belong to the same class. This enables us to classify new unseen images and calculate a distance between them and the training datasets.

A siamese convolutional neural network was trained for the comparison of different datasets, measuring the euclidean distance between the output vectors

$$d(\vec{u}, \vec{v}) = \|\vec{u} - \vec{v}\| = \sqrt{(v_1 - u_1)^2 + (v_2 - u_2)^2 \cdots (v_n - u_n)^2}. \quad (5)$$

The network is trained on pairs of images taken from datasets of different samples. During the training, the loss function is being minimized when similar morphology and signal distribution are found, e.g., both images belong to the same category or same biological sample. On the other hand, the loss function is being maximized if the images belong to different categories. For the training of the siamese network, we used a normalized fluorescent dataset of mice stem cell, *C. elegans* and *Drosophila melanogaster* with 15,000 images with a size of $64 \times 64$ for each biological specimen. We prepared the training dataset by pairing these image repositories of different biological models and setting a label of 1 for similar morphology, e.g., images paired to the same biological specimen (belonging to the same dataset) and 0 for images paired to a different specimen. During the training, we used contrastive loss with margin as the loss function

$$L_W = YD_W^2 + (1 - Y)\{\max(0, m - D_W)\}^2, \quad (6)$$

where $Y$ is the target of the network, 0 for dissimilar pairs and 1 for similar pairs, $D_W$ is the euclidean distance between the two inputs, and $m$ is the margin equal to 1. After the training, we compute the distance of each dataset compared to the zebrafish dataset. We saw a small distance (closer to 0) of the *Drosophila melanogaster* dataset against the zebrafish dataset, meaning that there is a strong correlation in the signal distribution. On the other hand, the distance between the zebrafish images against the *C. elegans* and stem cells images is high, which means a low similarity between these images (Supplementary Fig. 5).

**Calculation of the Signal/Noise ratio (SNR)**. To prove the improvement of the SNR after the restoration with the neural networks, we calculated the SNR of the input and the processed image. Here, we define the SNR as the ratio of the mean of the background subtracted signal intensity over the standard deviation of the background noise,

$$\text{SNR} = \frac{\mu_{\text{Signal}} - \mu_{\text{Background}}}{\sigma_{\text{Background}}}. \quad (7)$$

To select what was considered as noise and what was considered as signal, we sorted the intensities present in the image and calculate the gradient of the list of intensities. The cutting point was selected where a high inflection gradient was found in the distribution of intensities. The SNR computation only considered the pixels above that inflection point as signal and the rest as background.

**Quantifying spatial resolution**. To quantify the spatial resolution on images that were processed by a neural network, we employed a spatial resolution evaluator software by Fourier analysis[49]. The method breaks up the image into weighted sinusoidal functions, extracting the spatial frequency distribution of an image. Next, the radial mean of the power spectrum is taken and fitted to a rationalized function assuming a Gaussian point-spread function:

$$\log[I(k)] = A - Be^{-\frac{(k-k_0)^2}{2\sigma^2}} + Ce^{-\lambda|k-k_0|} + D|k - k_0|, \quad (8)$$

where $I(k)$ is the FT of the image, $A$, $B$, $C$, $D$, $k_0$, and $\lambda$ are constants that have been adjusted depending on the input (please refer to ref. [49]). The last three terms describe the azimuth averaged spectral content (AASC), the broadening of the peak function and the noise, respectively. This method is easy to use, and can be used to estimate the resolution of the structures present in the images based on the highest frequencies in the power spectrum. We performed the analysis on a distribution of $N = 10$ images randomly taken from the CARE dataset before and after restoration.

To estimate the resolving power of the light field microscope, we performed the same calculation after the deep-learning-based reconstruction and compared it to the ground-truth dataset. The images randomly taken from the body wall muscle and the *daf-16* dataset. Strictly, because these images do not contain diffraction-limited structures, the observation of the largest frequencies might not necessarily describe the resolution limit of the technique. However, they provide a conceptual comparison between the deep-learning-based image processing routines and the unprocessed ground truth.

**Statistics and reproducibility**. We choose the training, validation and testing datasets size to be representative of the variability seen across the timepoints. The models were optimized by based on a random validation set taken from the main dataset that had no overlap neither with the batch being processed for training nor the batch for testing. All training procedures were reproduced at least three times.

No statistical methods were applied to estimate a priori sample size. Technical replicates were defined as different images from the same treatment. Biological replicates were defined as different animals/cells. The statistical relation between normalized intensity and time in Fig. 2d was determined using R version 4.0.2 (2020-06-22). Statistical tests in Fig. 6c were performed in R, using a Wilcoxon test on ranked data. For Supplementary Fig. S2, procedures as outlined in ref. [80] were followed. In short, the effect sizes and CIs are reported above as: effect size [CI width-lower bound; upper bound]. The compute the paired median distribution,

5000 bootstrap samples were taken; the confidence interval is bias-corrected and accelerated. The *P*-value(s) reported are the likelihood(s) of observing the effect size(s), if the null hypothesis of zero difference is true. For each permutation *P*-value, 5000 reshuffles of the control and test labels were performed.

**Reporting summary**. Further information on research design is available in the Nature Portfolio Reporting Summary linked to this article.

## Data availability
All training datasets generated during and/or analyzed during the current study are available in the zenodo.org repository, https://doi.org/10.5281/zenodo.7018876. The source data are available as Supplementary Data 1.

## Code availability
The code of the training and inference pipelines and instructions to run it will be freely available under https://gitlab.icfo.net/NMSB/efmicro.

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

## Acknowledgements

We would like to thank the NMSB and SLN lab, especially Jordi Andilla for discussions and suggestion on optical design and imaging procedures. We thank Senda Jiménez-Delgado, Hanna-Maria Häkkinen, Queralt Tolosa and Neus Sanfeliu-Cerdan for help with molecular biology, worm and zebrafish maintenance and imaging. V.R. acknowledges financial support from the Ministerio de Ciencia y Innovacion through the Plan Nacional (PID2020-117011GB-I00) and funding from the European Union's Horizon EIC-ESMEA Pathfinder program under grant agreement No 101046620. M.K. acknowledges financial support from the ERC (MechanoSystems, 715243), HFSP (CDA00023/2018), Ministerio de Ciencia y Innovacion (PID2021-123812OB-I00 project funded by MCIN/AEI/10.13039/501100011033/FEDER, UE), FEDER (EQC2018-005048-P), "Severo Ochoa" program for Centres of Excellence in R&D (CEX2019-000910-S; RYC-2016-21062), from Fundació Privada Cellex, Fundació Mir-Puig, and from Generalitat de Catalunya through the CERCA and Research program (2017 SGR 1012), the Laserlab-Europe (H2020 GA no. 871124) in addition to funding through H2020 Marie Skłodowska-Curie Actions (754510 to A.G., and 847517 to L.F.M.C.).

## Author contributions

L.F.M.C., G.C., and M.K. built the microscope; L.F.M.C., A.C.G., M.P.R., L.L., V.V., and M.E.Q. performed experiments; L.F.M.C. wrote software and performed the deep learning; J.S., L.B., and M.P.R. performed transgenesis of mESCs and *C. elegans*; L.B., V.R., D.R., P.L.A., and M.K. supervised and M.K. conceived the project. L.F.M.C. and M.K. wrote the first draft, with input from all authors.

## Competing interests

The authors declare no competing interests.

## Additional information

 ns license, unless indicated otherwise in a credit line to the material. If material is not included in the article's Creative Commons license and your intended use is not permitted by statutory regulation or exceeds the permitted use, you will need to obtain permission directly from the copyright holder. To view a copy of this license, visit http://creativecommons.org/licenses/by/4.0/.

© The Author(s) 2022

