## [Peer Review File · Communications Biology]

Reviewers' comments:

Reviewer #1 (Remarks to the Author):

The study by Morales-Curiel et al developed a highly-efficiency microscope LowLiteScope to observe bioluminescent samples free of phototoxicity and combined it with a denoising technique. They used the CARE algorithm for denoising and successfully restored a clear bioluminescence image of a high signal-to-noise ratio from a degraded image of low intensity. They applied this technique for observation of neurons in *C. elegans* nematodes, plasma membrane in zebrafish embryo cells, and nuclei in mouse embryonic stem cells, and the denoising was proven to be useful for the segmentation of the nucleus. Furthermore, they introduced a microlens array into the LowLiteScope for 3D bioluminescence imaging by light-field microscopy. They used a CNN technique to devise an image reconstruction procedure involving denoising and deconvolution for the light-field image data to perform single-exposure volumetric bioluminescence microscopy. They used this technique to successfully restore 3D Ca²⁺ dynamics in freely-moving nematodes with an exposure time of 200-500 ms. To my impression, the experiment was conducted carefully, and the techniques and results are sound. However, I think that the manuscript should be more improved for clarity for the convenience of readers (see below).

Background information

In the introduction section, the authors well summarized the drawbacks of fluorescence and bioluminescence imaging. Furthermore, it would be better to briefly provide in the introduction or discussion the information of previous studies about denoising for bioluminescence imaging and 3D image reconstruction for light-field microscopy by machine-learning, if any. This information would help readers evaluate what the authors newly achieved.

LowLiteScope

The authors devised a microscope with an ultra-compact optical axis coupled to a single photon-resolving qCMOS camera, because they "were unable to observe any signal from Nanolanterns transfected into HeLa cells on a commercial compound microscope" (L. 42). By "a commercial compound microscope", they did not specify in the text what they used, but I guess that they meant Leica DMI8 inverted microscope. Although I have not tried Leica DMI8, I guess that its optical path would be carefully designed and the efficiency of light delivery to the camera port would not be very bad, like microscopes from other manufacturers. Moreover, they claimed, "with this new setup, we were able to obtain high SNR images for cells transfected with Nanolantern fusions ... for exposure times down to 2s" (L. 46). Thus, I think that LowLiteScope would be an important key to the present study and the readers are likely to be interested in how efficient the LowLiteScope was in comparison to a commercialized microscope. They should present a quantitative comparison between LowLiteScope and a commercial compound microscope.

How degraded a raw image can be for restoration?

The authors used the CARE algorithm to restore a high SNR image from a degraded image. The improvement of the restored images was impressive. Furthermore, the readers may wonder how badly degraded image can be failed (or successful) to be restored to a high SNR image. I hope that they could present some results concerning image quality with regards to things such as pixel-wise intensity (photons/pixel) of objects.

Teaching data

The text says, "prior knowledge of the underlying sample structure should facilitate superior image reconstruction with dramatically improved SNR using deep learning-based content aware image restoration (CARE) algorithms" (L. 76). Besides, they used "a pre-trained CARE model, originally established for epithelial monolayers in *Drosophila* wing discs" for denoising of zebrafish embryo cells. For the convenience of readers, they should mention criteria of what types of teaching data can be used for supervision.

L. 19

The text says, "they require chemicals with a poor bioavailability due to low solubility in water, which greatly limits the quantum yield and concomitant signal-to-noise ratio (SNR)." Generally, a quantum yield in this context is a probability of the emission of luminescence photon per luciferase

turnover. The low solubility of luciferase substrate in water may limit the SNR, but the quantum yield would not be related to the low solubility in water.

L. 106

The text says, "with a small, but diversified training dataset". However, up to this point in the text, they did not show the data size.

L. 112

I guess that "one photon live-cell imaging" means one photon excitation live-cell imaging. If I am correct, "excitation" had better be inserted for clarity.

L. 114

About the sentence "We thus compared the activation of...", I guess that, chronologically, firstly they generated transgenic animals, and secondly, they compared the activation. It would be better to describe this in a way so that the chronological order can easily be understood by readers.

L. 116:

About "reporter for various stresses, including reactive oxygen species", its specific name had better be provided here, if possible. Because they wrote "both reporters" (L. 120), I guess that "both reporters" corresponded to "mNeonGreen-NanoLantern fused to DAF-16" and a "reporter for various stresses".

L. 118

If possible, a reference should be cited for "cytoplasmic/nuclear shuttling".

L. 121

The text says, "the activity of the bioluminescent reporter was more pronounced." However, many readers may not know well what happens to DAF-16 under stress, and therefore, it may not be easy for us to understand what was "more pronounced", by just looking at Figure 2. The authors should explicitly explain what pattern and what change in Figure 2 the readers are supposed to notice to understand that "the bioluminescent reporter was more pronounced".

In addition, to my understanding, they tried to use two types of stress, heat and excitation light in this experiment. Thus, for example, the experiment using "animals that were recorded with fluorescence microscopy" implies that they operationally applied light stress to the animals to evaluate its effect. I hope that the authors could clarify this point in the text.

Additionally, it is advisable to check if the notation of values such as "1e-5" suits the guideline of the journal.

L. 150

The authors claimed, "Expectedly, the images were noisy, due to the low quantum yield." They mentioned solubility of the substrate, catalytic turnover, and quantum yield in reference to properties of bioluminescent probes in L. 13-21. They also suggested that the expression level of mTurquoise-NL could also affect the brightness. Thus, I wonder how they found that the noisy images were attributed to "the low quantum yield".

L. 193

The authors claimed that they created "a full 3D stack of the bioluminescent scene which corresponded to a z-resolution of 1.5 μm and reconstructed 31 z-planes in conventional widefield microscopy". I think that they need to clarify the meaning of "a z-resolution of 1.5 μm ": just a step size in the z direction for the reconstructed 31 z-planes, or things such as FWHM of the point-spread function in the x-z plane. Moreover, it would be better to present a plot of point-spread functions in this method on the x-y and x-z planes.

L. 387

For "a U-Net type architecture (Denoising CARE 2D topology) for training and validation", a paper should be cited, if applicable.

Typos and small comments:

L. 7: For the abbreviation of "exempli gratia", it is more common to write "e.g."

L. 71: "in" might be missing before "living animas that express...". Besides, this sentence should be written in the past simple, because doing the experiment and obtaining the result were events that happened in the past. Similar misuse of present simple tense is seen somewhere else in the text.

L.83: "hogh"

L. 91: In "configuration16, 26)", delete ")"

L. 91: "training dataset (Fig. 1, 1d ii)" should be "training dataset (Fig. 1d ii)".

L. 101: When I read "(Fig. 1h, 24, 27)" literally, I thought that Fig. 1h was from refs 24 and 27. But I do not think that this was what the authors meant. Looking at the comma (,) that follows "1h", I guess that they might have deleted something that cited refs 24 and 27.

L. 127: "if"

L. 135: The definition of "EVL" should be provided somewhere in the text.

L. 302: In "Hamamatsu Orca Quest, or Fusion" and somewhere else in the text, that camera manufacturer is actually called Hamamatsu Photonics.

L.321: "Lowlight microscope" probably means "LowLiteScope".

L. 312: I guess that "100 mm" would be the focal lengths of the lenses. Anyway, the authors should specify what "100 mm" is about.

L. 373: The dichroic mirror FF459/526/596-Di01-25x36 is a product of Semrock Co.

L. 410: "µm" should be corrected for a unit of volume.

L. 477: "CMOS board level camera C11440-62U" appears here first. It is a product of Hamamatsu Photonics.

Reviewer #2 (Remarks to the Author):

This is an interesting manuscript that attempts to improve the practicality of bioluminescence imaging by incorporating deep learning methods. Traditionally, the strengths of bioluminescence relative to fluorescence are its 'background free' nature and low photodamage. The main weakness of bioluminescence is poor SNR, due to the extremely low light levels emitted by the sample. Krieg and coworkers attempt to address SNR by employing deep learning based denoising pipelines to restore the noisy bioluminescence data, using ground truth gleaned from higher SNR fluorescence data. They also combine denoising with a deep learning method for light-field reconstruction, showing that the combination of networks enables bioluminescence light field imaging. Applications are carefully chosen across a range of model systems and – while the image quality is still relatively modest – the authors have convinced me that indeed there is benefit in combining state-of-the-art computational methods with bioluminescent imaging. More generally, I think it is important to make progress in the area of ultralow light imaging, especially given the (probably underappreciated) role that phototoxicity in live cell imaging. I have a series of suggestions – mostly to do with the clarity and writing of the manuscript – that I would like to see addressed before recommending publication.

-Another drawback of bioluminescence imaging, as opposed to fluorescence imaging, is the lack of optical sectioning in the former. Can the authors comment on this issue, and whether deep learning effectively improves sectioning compared to the raw data?

-The spatial resolution of the technique remains poor, even after deep learning. It would be valuable to quantify spatial resolution in a few cases, relative to the input bioluminescence imaging as well as the ground truth fluorescence data. The authors mention 'diffraction-limited' in their paper – but do they actually achieve diffraction-limited performance? Please either demonstrate this assertion or remove it. It would also be important to comment in the light-field case if the resolution of the deep learning prediction is on par with the fluorescence ground truth.

-Along the same lines, it would be helpful for the authors to provide concrete examples of failures of their method. Is the method suitable for subcellular imaging, or is there a spatial resolution limit beyond which the deep learning methods cannot enhance resolution?

-The authors mention the term 'SNR' multiple times, as well as 'poor SNR', 'high SNR', and

'effective SNR'. It would be good to define what they mean on a more quantitative basis, especially since the key benefit of their method is that SNR is increased. How do they define SNR, by what factor does it increase, and how does the SNR of the deep learning compare to the fluorescence ground truth used to train their deep learning methods?

-The light field (re)training method with synthetic and experimental data is a bit difficult to follow, and I do not find the schematic in Fig. 4b very helpful. I would suggest a supplementary figure with a detailed, step-by-step workflow so the interested reader can follow exactly what was done here.

Minor comments

-First paragraph introduction – don't even the most recent versions of Nanolantern require a cofactor which is rate limiting, effectively limiting the emission rate far below that of fluorescence? It would be good to clarify – for the probes used in this paper – what the real limitation is. Is it poor solubility of the cofactor or the fact that the turnover rate is limited by diffusional collision with the cofactor? Or both?

-The authors mention 'novel cofactor chemistry', implying that they have done something novel here. Is this true? If so, please explain. If the work has been done before, is it still novel?

-what is the 'q' in 'qCMOS'? Quantitative? I would define this when first using the abbreviation.

-line 82, 'in-existent' -> 'non-existent'?

-line 105 'in living animals and tissue culture'. Has the application of deep learning to tissue culture been demonstrated at this point in the paper? This phrase makes it sound like this has, but as far as I can tell Fig. 1b is a demonstration that luminescence can be directly observed in tissue culture (i.e., without deep learning).

-It is worth emphasizing in the caption to Fig. 1b that the magnification level for the fluorescence vs. bioluminescence is different (i.e., list the mag values in the caption)

-line 103 'diffraction-limited axon caliber' – is the image actually diffraction-limited, i.e., does the axon actually appear to have diffraction-limited width? Mentioning the value for the apparent width would be useful to verify this assertion.

-line 111: 'Photobleaching during fluorescence microscopy is an indicator for potential photo damage...' I agree with this statement but am not clear that the results in this paragraph actually demonstrate this point. Was there substantial photobleaching in the fluorescence data?

-line 112: 'wavelength' -> 'wavelengths'

-line 122: 'more pronounced'. Than what? The fluorescence stress reporter? Indicate this explicitly.

-line 137 – 'strong out of focus haze' with bioluminescence. Presumably this does not occur in fluorescence imaging because a confocal system with optical sectioning was used – if this is the case, please state this point clearly in the main text.

-I am unclear from reading the text what experiments did 'conventional' bioluminescence Z-stacking, as opposed to light-field imaging. Or was this done with the mESC cell tracking? Please clarify in which samples z stacks were taken and in which samples only 2D imaging was performed.

- 'millisecond exposures' in abstract is somewhat misleading – nothing in this paper was imaged at the 'millisecond' timescale and even the light-field recordings actually were considerably longer

-The authors mention in line 192, 'z resolution of 1.5 μm ' – is this really true? The z resolution looks pretty poor in this figure

-schematic in Fig. 4a too small to be usefully read

-Scale bars on Fig. 4b, d 4f, 4g?

-line 484, 'apply an image registration' – please provide more details, what sort of image registration, and with what program/method?

-line 493, 'rectifying the light field image', what is meant by this?

-What are the values on the colorbars on Supplementary Figs. 1b, c? It is very difficult to understand what we are looking at here and how to interpret colors. I would assume warmer colors on the SSIM mean better, and cooler colors for RSE. Please clarify.

-The majority of the images in the SI are missing scale bars, and sometimes even when scale bars are given, there is no value in the caption (e.g., SI Fig. 2)

-Scale bar values for SI videos?

-In SI Video 2 (spheroid), there appear to be large fluctuations in the signal from one frame to the next. Is this because of focal drift? Please comment.

We thank the two anonymous reviewers for their valuable input. We strongly feel that their questions and suggestions significantly improved the presentation of the method and guided the evaluation of the deep learning procedures. We have addressed all the comments as outlined below with specific emphasis on evaluating the signal/noise ratio, the resolution limits of the deep learning base restoration and light field reconstructions and the general limitations of the procedures. We hope that the referees share our point of view.

1 Reviewer #1 (Remarks to the Author)

The study by Morales-Curiel et al developed a highly-efficiency microscope LowLiteScope to observe bioluminescent samples free of phototoxicity and combined it with a denoising technique. They used the CARE algorithm for denoising and successfully restored a clear bioluminescence image of a high signal-to-noise ratio from a degraded image of low intensity. They applied this technique for observation of neurons in *C. elegans* nematodes, plasma membrane in zebrafish embryo cells, and nuclei in mouse embryonic stem cells, and the denoising was proven to be useful for the segmentation of the nucleus. Furthermore, they introduced a microlens array into the LowLiteScope for 3D bioluminescence imaging by light-field microscopy. They used a CNN technique to devise an image reconstruction procedure involving denoising and deconvolution for the light-field image data to perform single-exposure volumetric bioluminescence microscopy. They used this technique to successfully restore 3D Ca²⁺ dynamics in freely-moving nematodes with an exposure time of 200-500 ms. To my impression, the experiment was conducted carefully, and the techniques and results are sound. However, I think that the manuscript should be more improved for clarity for the convenience of readers (see below).

1. Background information

In the introduction section, the authors well summarized the drawbacks of fluorescence and bioluminescence imaging. Furthermore, it would be better to briefly provide in the introduction or discussion the information of previous studies about denoising for bioluminescence imaging and 3D image reconstruction for light-field microscopy by machine-learning, if any. This information would help readers evaluate what the authors newly achieved.

We have now included a detailed introduction about denoising in ultra low-light imaging and 3D light field reconstruction of fluorescence images from plenoptic lightfield images using deep learning tools.

2. LowLiteScope

The authors devised a microscope with an ultra-compact optical axis coupled to a single photon-resolving qCMOS camera, because they “were unable to observe any signal from Nanolanterns transfected into HeLa cells on a commercial compound microscope” (L. 42). By “a commercial compound microscope”, they did not specify in the text what they used, but I guess that they meant Leica DMI8 inverted microscope. Although I have not tried Leica DMI8, I guess that its optical path would be carefully designed and the efficiency of light delivery to the camera port would not be very bad, like microscopes from other manufacturers. Moreover, they claimed, “with this new setup, we were able to obtain high SNR images for cells transfected with Nanolantern fusions . . . for exposure times down to 2s” (L. 46). Thus, I think that LowLiteScope would be an important key to the present study and the readers are likely to be interested in how efficient the LowLiteScope was in comparison to a commercialized microscope. They should present a quantitative comparison between LowLiteScope and a commercial compound microscope.

The reviewer brings up an important point. We have now provided a direct comparison of bioluminescence using the same sample acquired from a commercial microscope and the optimized low-light microscope. We included this as a new figure S1 for a cultured cells and worms. In particular we write, starting on line 145: “Indeed, we only observed a weak signal from Nanolanterns transfected into HeLa cells or transgenic *C. elegans* on a commercial compound microscope at the maximum exposure time of our camera (Supplementary Figure 1).”

3. How degraded a raw image can be for restoration?

The authors used the CARE algorithm to restore a high SNR image from a degraded image. The improvement of the restored images was impressive. Furthermore, the readers may wonder how badly degraded image can be failed (or successful) to be restored to a high SNR image. I hope that they could present some results concerning image quality with regards to things such as pixel-wise intensity (photons/pixel) of objects.

We thank the reviewer for this suggestion. To address this point we first have calculated the SNR for all images in the main text figures before and after denoising to visually underline the difference. Next, we have calculated the SNR of the input images and the corresponding quality metric of the prediction. To demonstrate this to the readers, we have now included a new supplementary figure 3 that indicated the structural similarity as a function of the SNR of the input picture. It can be seen that input pictures with a higher SNR achieve a better outcome from the inferences than pictures with a lower SNR. With all due respect for the reviewers suggestion, we have refrained from quantifying the pixel-wise intensity as a cutoff metric, as especially from thick samples, out-of-focus light adds significantly to the signal (but also to the noise) - thus, the quantification of the pixel intensity would lead to the wrong perception that large photon counts can lead to weak predictions. We have now included this description starting on line 239, line 353, and Supplementary Figure 3 and 8.

4. Teaching data

The text says, “prior knowledge of the underlying sample structure should facilitate superior image reconstruction with dramatically improved SNR using deep learning-based content aware image restoration (CARE) algorithms” (L. 76). Besides, they used “a pre-trained CARE model, originally established for epithelial monolayers in *Drosophila* wing discs” for denoising of zebrafish embryo cells. For the convenience of readers, they should mention criteria of what types of teaching data can be used for supervision.

This is a great question of the referee and a common issue in deep learning methods. In short, we reasoned that the underlying geometry and signal distribution of the two tissues under question are similar enough to generalize the denoising network such that it would perform well on both. To facilitate the answer quantitatively, we have now compared the sample structure and geometry between zebrafish epithelial layer and the *Drosophila* wing disc labeled at the membrane using a Siamese neuronal network. This network compares two input images and measures the similarity of both images by means of a Euclidean distance metric (see Methods for details and ref Koch et al, 2015). The lower the Euclidean distance (and closer to zero), the more similar the two input images. As can be seen in the new Supp. Fig. S5 we find that the distribution of fluorescent marker is very similar in both structures, a reason for why the pre-trained model can perform well with this type of data. In contrast, the model trained for *C. elegans* muscles does not perform and yields a large distance due to a different morphology and intensity distribution of fluorescent markers. We have now explained this in the text and provided a new Fig. S5 to justify the use of the *Drosophila* model to reconstruct zebrafish embryos. In addition, we also used the same method to quantitatively assess that we can use our muscle-trained *C. elegans* model with high confidence on images depicting neuronal structures. The description is now included in Supplementary Figure 5 and line 296.

5. L. 19

The text says, “they require chemicals with a poor bioavailability due to low solubility in water, which greatly limits the quantum yield and concomitant signal-to-noise ratio (SNR).” Generally, a quantum yield in this context is a probability of the emission of luminescence photon per luciferase turnover. The low solubility of luciferase substrate in water may limit the SNR, but the quantum yield would not be related to the low solubility in water.

We thank the referee for pointing this out. We have now clarified this issue and the distinction in the text by rewriting the introduction.

6. L. 106

The text says, “with a small, but diversified training dataset”. However, up to this point in the text, they did not show the data size.

We have now included what precisely we mean by the size of the data set and how it compares to other deep learning pipelines. Specifically we now write “with a small, but diversified training dataset composed of 2500 different image pairs”. This value is on the upper end for bioimage training data sets and as many as 3000 images have been used by Wang et al. [1] and as little as 40 have been used by Ounkomol et al.[2]. See line 272 and the Methods section with a detailed description of the training procedure.

7. L. 112

I guess that “one photon live-cell imaging” means one photon excitation live-cell imaging. If I am correct, “excitation” had better be inserted for clarity.

Thank you.

8. L. 114

About the sentence “We thus compared the activation of . . .”, I guess that, chronologically, firstly they generated transgenic animals, and secondly, they compared the activation. It would be better to describe this in a way so that the chronological order can easily be understood by readers.

We have now clarified the sequence of event and the chronological order of the experiment. Given the various comments below, we rewrote the entire paragraph and included the DAF-16 activity as a separate Fig. 2, which we introduce before the deep learning explanation. See section starting at line 165.

9. L. 116:

About “reporter for various stresses, including reactive oxygen species”, its specific name had better be provided here, if possible. Because they wrote “both reporters” (L. 120), I guess that “both reporters” corresponded to “mNeonGreen-NanoLantern fused to DAF-16” and a “reporter for various stresses”.

We have now clarified this issue in the main text. See section starting at line 165.

10. L. 118

If possible, a reference should be cited for “cytoplasmic/nuclear shuttling”.

We have now cited the paper by Hsu et al “Regulation of Aging and Age-Related Disease by DAF-16 and Heat-Shock Factor” at this point to introduce the fact that daf-16 is activated by heat and translocates into the nucleus. See section starting at line 165.

11. L. 121

The text says, “the activity of the bioluminescent reporter was more pronounced.” However, many readers may not know well what happens to DAF-16 under stress, and therefore, it may not be easy for us to understand what was “more pronounced”, by just looking at Figure 2. The authors should explicitly explain what pattern and what change in Figure 2 the readers are supposed to notice to understand that “the bioluminescent reporter was more pronounced”.

We thank the referee for this suggestion. We have now described in greater detail what DAF-16 is and how it is related to stress. See section starting at line 165.

In addition, to my understanding, they tried to use two types of stress, heat and excitation light in this experiment. Thus, for example, the experiment using “animals that were recorded with fluorescence microscopy” implies that they operationally applied light stress to the animals to evaluate its effect. I hope that the authors could clarify this point in the text.

We thank the referee for pointing out the flaws in this paragraph. We have rewritten the entire section to make the

statements clear and more precise. See section starting at line 165.

Additionally, it is advisable to check if the notation of values such as “1e-5” suits the guideline of the journal.

12. L. 15

The authors claimed, “Expectedly, the images were noisy, due to the low quantum yield.” They mentioned solubility of the substrate, catalytic turnover, and quantum yield in reference to properties of bioluminescent probes in L. 13-21. They also suggested that the expression level of mTurquoise-NL could also affect the brightness. Thus, I wonder how they found that the noisy images were attributed to “the low quantum yield”.

We have removed this statement from the text, as it is irrelevant for the flow of arguments. We agree that we do not have the information about all the factors that limit the signal/noise ratio.

13. L. 193

The authors claimed that they created “a full 3D stack of the bioluminescent scene which corresponded to a z-resolution of 1.5 μm and reconstructed 31 z-planes in conventional widefield microscopy”. I think that they need to clarify the meaning of “a z-resolution of 1.5 μm ”: just a step size in the z direction for the reconstructed 31 z-planes, or things such as FWHM of the point-spread function in the x-z plane. Moreover, it would be better to present a plot of point-spread functions in this method on the x-y and x-z planes.

We have now clarified this sentence in the text and suggest that our sampling in the z direction for the reconstructed 31 z-planes corresponds to 1.5 μm . Specifically, we write “we were able to obtain three dimensional calcium recordings from whole animals with typically 200-500ms exposure time per light field image, which we reconstructed to create a full 3D stack of the bioluminescent scene represented by 31 z-planes with a spacing of 1.5 μm between sequential planes”. In addition, we have added the complete characterization of the spatial resolution in the deep learning pipelines using a Fourier transform of the input, ground truth and predicted images to measure the largest observable frequencies in the power spectral density (refer to Figure 3 and S11, following the procedure described in [3]). As we and others [4] noted, the spatial resolution after deep learning light field reconstruction is not bound by the point spread function of the lenslet array and can yield significantly better resolution than what is expected from diffraction optics. The underlying reason for this is that the model for lightfield image reconstruction was trained on ground truth data derived from conventional fluorescence microscopy z-stacks that has a nominal resolution close to the diffraction limit. We have added this data as a new supplementary Figure S11 and discuss these results and their implications in the Discussion.

14. L. 387

For “a U-Net type architecture (Denoising CARE 2D topology) for training and validation”, a paper should be cited, if applicable. We have now cited the literature at this point as well.

15. Typos and small comments:

L. 7: For the abbreviation of “exempli gratia”, it is more common to write “e.g.”

L. 71: “in” might be missing before “living animas that express...”. Besides, this sentence should be written in the past simple, because doing the experiment and obtaining the result were events that happened in the past. Similar misuse of present simple tense is seen somewhere else in the text.

L.83: “hogh”

L. 91: In “configuration16, 26)”, delete “)”

L. 91: “training dataset (Fig. 1, 1d ii)” should be “training dataset (Fig. 1d ii)”.

L. 101: When I read “(Fig. 1h, 24, 27)” literally, I thought that Fig. 1h was from refs 24 and 27. But I do not think that this was what the authors meant. Looking at the comma (,) that follows “1h”, I guess that they might have deleted

something that cited refs 24 and 27.

L. 127: “if”

L. 135: The definition of “EVL” should be provided somewhere in the text.

L. 302: In “Hamamatsu Orca Quest, or Fusion” and somewhere else in the text, that camera manufacturer is actually called Hamamatsu Photonics.

L.321: “Lowlight microscope” probably means “LowLiteScope”.

L. 312: I guess that “100 mm” would be the focal lengths of the lenses. Anyway, the authors should specify what “100 mm” is about.

L. 373: The dichroic mirror FF459/526/596-Di01-25x36 is a product of Semrock Co.

L. 410: “m” should be corrected for a unit of volume.

L. 477: “CMOS board level camera C11440-62U” appears here first. It is a product of Hamamatsu Photonics.

We are truly grateful for the reviewer for helping us to find these typos and improving the wording.

2 Reviewer #2 (Remarks to the Author):

This is an interesting manuscript that attempts to improve the practicality of bioluminescence imaging by incorporating deep learning methods. Traditionally, the strengths of bioluminescence relative to fluorescence are its ‘background free’ nature and low photodamage. The main weakness of bioluminescence is poor SNR, due to the extremely low light levels emitted by the sample. Krieg and coworkers attempt to address SNR by employing deep learning based denoising pipelines to restore the noisy bioluminescence data, using ground truth gleaned from higher SNR fluorescence data. They also combine denoising with a deep learning method for light-field reconstruction, showing that the combination of networks enables bioluminescence light field imaging. Applications are carefully chosen across a range of model systems and – while the image quality is still relatively modest – the authors have convinced me that indeed there is benefit in combining state-of-the-art computational methods with bioluminescent imaging. More generally, I think it is important to make progress in the area of ultralow light imaging, especially given the (probably underappreciated) role that phototoxicity in live cell imaging. I have a series of suggestions – mostly to do with the clarity and writing of the manuscript – that I would like to see addressed before recommending publication.

1. Another drawback of bioluminescence imaging, as opposed to fluorescence imaging, is the lack of optical sectioning in the former. Can the authors comment on this issue, and whether deep learning effectively improves sectioning compared to the raw data? We thank the reviewer for this suggestion. The lack of optical sectioning in the bioluminescence technique leads to a lower SNR and contrast especially in thick samples such as the zebrafish embryo, because the whole sample constitutively emits light and not only the plane that gets excited in the focus. This is in contrast to confocal microscopy, which occludes out-of-focus light through a pinhole aperture. However, for thin samples this effect is less dramatic. We have now commented on this issue in the results in relation to the zebrafish imaging and in the discussion. Specifically we write starting on line 287: “Because this out-of-focus background is absent in the fluorescent images due to the optical sectioning of the confocal microscope, we reasoned that the blurred bioluminescent images can be reconstructed using a neuronal network trained on clean images derived from background-free confocal microscopy. ... Despite the challenging task due to the poor input SNR, we found that this model was generalizable (Fig. S5) and fit our input extremely well, being able to greatly improve the SNR (Fig. 5b,c) and remove the blur that originated from out-of-focus. Critically, these signal restorations and improvements enabled the segmentation of individual cells in the embryo (Fig. 5c) which may afford the calculation of their perimeter and cell area and possibly increases the optical sectioning capability of the bioluminescence microscopy - a procedure that otherwise would not

be possible.” In addition, we also discussed the potential use of dedicated machine learning procedures to increase z-sectioning in widefield microscopy data [5]. See also line 307.

2. The spatial resolution of the technique remains poor, even after deep learning. It would be valuable to quantify spatial resolution in a few cases, relative to the input bioluminescence imaging as well as the ground truth fluorescence data. The authors mention ‘diffraction-limited’ in their paper – but do they actually achieve diffraction-limited performance? Please either demonstrate this assertion or remove it. We have now removed the statement to avoid confusion with diffraction limited resolution, and restricted our description to the diffraction-limited size of the observed neurons. We have also quantified the gain in spatial resolution after deep learning denoising by quantifying the largest frequencies in the Fourier domain (according to [3]) and added these results into the new Figure 3. It is clearly visible by comparing the black line with the blue lines that the deep learning procedure increases the resolution close to the expected value of the synthetic PSF and the ground truth. We noted though, that the denoising of the images depicting the neuronal morphologies appeared to broaden the width of the axonal caliper, which we attributed to the fact that we used a neuronal network trained on muscle images that often contain wider intensity distributions. We have discussed this in the main text lines 223 and onwards and the entire Discussion section.

It would also be important to comment in the light-field case if the resolution of the deep learning prediction is on par with the fluorescence ground truth.

See also comment 13 from Reviewer 1. We have performed a similar analysis using the reconstructed light field images and compared the maximum recovered frequencies with the ground truth from fluorescence and bioluminescence images in the power spectral densities after performing a FFT on the reconstructed and rectified ground truth images that we used for the training process and found that they are indeed comparable. The results are very interesting and we added the accompanying data in the Supplementary Figure 11. From classical diffraction optics of the microlens array we expected a nominal resolution of $11.1\ \mu\text{m}$ which we derived dividing the pitch with the objective magnification. As can be seen from the quantification in Figure S11, our largest frequencies correspond to a distance of $2\ \mu\text{m}$, approximately ≈ 5 fold increase in spatial resolution. This essentially confirmed the results from [4], that the deep learning based light field reconstruction performs better than the classical deconvolution, as the network learns to reconstruct the light field image with the conventional z-stack that has the nominal resolution of a widefield optical microscope. However, because the ground truth images were synthetically downsampled [6] to match the virtual pixels for each lenslet of the microlens array, this rectification led to a reduction in spatial resolution of the rectified images compared to the unprocessed ground truth (see Fig. S11a). Due to this, the spatial resolution still lacks behind the unprocessed ground truth, which we determined to be $\approx 0.4\ \mu\text{m}$ (Fig. S11). We discussed this extensively in the main text.

3. Along the same lines, it would be helpful for the authors to provide concrete examples of failures of their method. We thank the referee for this comment and have included several specific drawbacks and failures of our method that we experienced through out the development of this project. We enumerated them in the discussion and throughout the text. As pointed out above, the reconstructions fail or are unsatisfactory when the input SNR is too low. The method also fails if the network has not been trained properly or the underlying sample structure deviates significantly from the signal distribution in the ground truth data. We have shown this specifically in various display items, e.g. Supplementary Fig 3. We also failed to create a dedicated model using supervised training pipelines to specifically restore image data derived from neurons. The reason for this failure is that the low expression of the fluorophores in the neuronal transgenics results in a large contribution of the autofluorescence in the training images (much higher than in the muscle data), which affects the inference quality. The result is that the particular, dedicated models learn to recognize and ‘imagine’ autofluorescence in the bioluminescence images which creates spurious signals and artefacts. That being said, we found that the muscle-trained model is largely generalizable and can infer high-SNR images from

low quality images depicting neurons. We have enumerated specific failures that we encountered in the discussion and in appropriate occasions throughout the text. The discussion starts in line 391

4. Is the method suitable for subcellular imaging, or is there a spatial resolution limit beyond which the deep learning methods cannot enhance resolution? In principle, the same resolution as in fluorescence microscopy could be achieved with bioluminescence and we quantified the resolution limit in Fig. 3. We also showed that bioluminescence microscopy is poised to resolve subcellular dynamics, as shown for nuclear trafficking of DAF-16 (Fig. 2) and actin fibers in cultured HeLa cells (Fig. 1), among others. Using appropriate training data from superresolved images, it might even be possible to ‘break’ the diffraction barrier. However, light field imaging is unable to resolve subcellular features in small *C. elegans* cells.
5. The authors mention the term ‘SNR’ multiple times, as well as ‘poor SNR’, ‘high SNR’, and ‘effective SNR’. It would be good to define what they mean on a more quantitative basis, especially since the key benefit of their method is that SNR is increased. How do they define SNR, by what factor does it increase, and how does the SNR of the deep learning compare to the fluorescence ground truth used to train their deep learning methods?

We thank the referee for bringing up this important point. We define SNR as the ratio of the mean of the background subtracted signal intensity over the standard deviation of the background noise ($SNR = \frac{\mu_{Signal} - \mu_{Background}}{\sigma_{Background}}$). We choose a predefined ROI that clearly does not contain any sample structure and is only composed of background pixels. The ‘signal’ in the background pixels partitions into dark noise, read out noise and photons collected due to sample noise (autofluorescence, out of focus blur, etc). To facilitate a direct comparison between the degraded input images and clean CARE predictions, we have now added the SNR for each image that we show in the main figure, whenever a quantitative comparison became relevant, so that the values can directly be parsed from the figures. Since the performance and increase in SNR does not follow a specified factor and is very variable based on input SNR etc, we believe a population average is not helpful to describe the methodological advance. However, in general we found a 5-10fold increase in SNR after denoising.

6. The light field (re)training method with synthetic and experimental data is a bit difficult to follow, and I do not find the schematic in Fig. 4b very helpful. I would suggest a supplementary figure with a detailed, step-by-step workflow so the interested reader can follow exactly what was done here.

We now provide the suggested supplementary figure (Fig. S7) for the reconstruction work flow. We hope that the reviewer appreciates this schematic and facilitates the understanding of our procedure.

Minor comments

7. First paragraph introduction – don’t even the most recent versions of Nanolantern require a cofactor which is rate limiting, effectively limiting the emission rate far below that of fluorescence? It would be good to clarify – for the probes used in this paper – what the real limitation is. Is it poor solubility of the cofactor or the fact that the turnover rate is limited by diffusional collision with the cofactor? Or both? We have provided more context and clarification on the limiting factors for photon production in a luciferase reaction. In short, we agree with the referee that the rate limiting step is the enzymatic turnover. We thank the reviewer for pointing out this statement. Because the same probe provide images with a high signal/noise ratio upon direct excitation of the mTurquoise fluorescent proteins by 430nm LED light at shorter camera exposure times, we reasoned that the photophysical properties of luminescence and fluorescence emission is strongly different. Indeed, Steinmeyer estimated that the photon emission of a yellow fluorescent protein at its saturation excitation power is ≈ 3000 photons/ms[7]. In contrast, enhanced nanolanters were shown to emit 10photon/s[8]. We have clarified this in the text.

8. The authors mention ‘novel cofactor chemistry’, implying that they have done something novel here. Is this true? If so, please explain. If the work has been done before, is it still novel?
We have now rewritten the introductory paragraph to avoid any confusion about novelty and approaches undertaken to improve the bioluminescence microscopy.
9. what is the ‘q’ in ‘qCMOS’? Quantitative? I would define this when first using the abbreviation. Done
10. line 82, ‘in-existent’ -> ‘non-existent’? Done
11. line 105 ‘in living animals and tissue culture’. Has the application of deep learning to tissue culture been demonstrated at this point in the paper? This phrase makes it sound like this has, but as far as I can tell Fig. 1b is a demonstration that luminescence can be directly observed in tissue culture (i.e., without deep learning).
Yes, the deep learning has been already introduced at this point in the text. To avoid confusion, we separated Fig. 1 into three separate display items. Now, Fig. 1 and 2 contains only unprocessed bioluminescent images, whereas Fig. 3 and 4 introduces the reader to the CARE algorithms and training pipelines used for denoising.
12. It is worth emphasizing in the caption to Fig. 1b that the magnification level for the fluorescence vs. bioluminescence is different (i.e., list the mag values in the caption)
Thank you, we have included the nominal magnification values of the imaging system in the caption
13. line 103 ‘diffraction-limited axon caliper’ – is the image actually diffraction-limited, i.e., does the axon actually appear to have diffraction-limited width? Mentioning the value for the apparent width would be useful to verify this assertion.
We have now provided factual values and a reference to direct the reader to electron microscopy images of C elegans neurons, including ASH. Even though the axon has a diameter that approaches the resolution limit of a light microscope, the images are likely not approaching this value. The underlying reason is the low signal in the input image and the widening of the axon diameter by the deep learning model (which was developed on muscle cells and thus likely applies this knowledge also on neurons).
14. line 111: ‘Photobleaching during fluorescence microscopy is an indicator for potential photo damage...’ I agree with this statement but am not clear that the results in this paragraph actually demonstrate this point. Was there substantial photobleaching in the fluorescence data?
We have clarified this statement.
15. line 112: ‘wavelength’ – > ‘wavelengths’ Done
16. line 122: ‘more pronounced’. Than what? The fluorescence stress reporter? Indicate this explicitly. Done
17. line 137 – ‘strong out of focus haze’ with bioluminescence. Presumably this does not occur in fluorescence imaging because a confocal system with optical sectioning was used – if this is the case, please state this point clearly in the main text.
Also related to comment 1. We have now rewritten this paragraph to better explain the experiment and analyses. Specifically, we write: ‘strong out-of-focus haze obscured the signal strength and SNR in bioluminescence contrast. Despite this challenging condition, we were still able to record a signal reminiscent of the cell junctions after 100ms exposure time (Fig. 4b, Supplementary Fig. S4), however, post-processing such as segmentation proved difficult due to the poor SNR between the specific membrane label and unspecific background. Because this out-of-focus background is absent in the fluorescent images due to the optical sectioning of the confocal microscope, we reasoned that the blurred bioluminescent images can be reconstructed using a neuronal network trained on clean images derived from background-free confocal microscopy. Unfortunately, due to tissue movements during the acquisition time in the point

scanning confocal microscope, we were not able to collect paired image stacks (degraded vs ground truth) to assemble a dedicated training pipeline. We thus processed the bioluminescent images with a pre-trained CARE model, originally established to reconstruct low SNR images from epithelial monolayers in *Drosophila* wing discs[9], a tissue with similar tessellated morphology and signal distribution.'

18. I am unclear from reading the text what experiments did 'conventional' bioluminescence Z-stacking, as opposed to light-field imaging. Or was this done with the mESC cell tracking? Please clarify in which samples z stacks were taken and in which samples only 2D imaging was performed. We apologize for the confusion. We have now clarified this issue throughout the text and called out where conventional z-stacks were recorded as opposed to plenoptic light field images. For the mESC cell tracking, no z-stack or lightfield images were acquired.

19. 'millisecond exposures' in abstract is somewhat misleading – nothing in this paper was imaged at the 'millisecond' timescale and even the light-field recordings actually were considerably longer
We have now changed the word to 'sub-second' instead of millisecond.

20. The authors mention in line 192, 'z resolution of 1.5 μm ' – is this really true? The z resolution looks pretty poor in this figure

Also related to comment 13 of reviewer 1: We have rephrased the sentence to reflect what we actually meant to say - a spacing of 1.5 μm between two consecutive frames within the z-stack.

21. schematic in Fig. 4a too small to be usefully read Done

22. Scale bars on Fig. 4b, d 4f, 4g?

Done

23. line 484, 'apply an image registration' – please provide more details, what sort of image registration, and with what program/method?

We have added these details to the methods section. In short, we used `pystackreg`, a python library to register two or more images to correct for different distortions (translation, rotation, scaling, shearing and bilinear transformation). The one we used was the scaled rotation, which does translation + rotation + scaling to find the correct transformation matrix for the alignment. We have added this information to the methods.

24. line 493, 'rectifying the light field image', what is meant by this? The rectification is performed on both, the light field images and the 3D ground truth image stack to ensure that the pixels perfectly correlate to each other such that the information content in the one is matching the information in the other. We have added this information to the methods section. The rectification on the light field images is a resampling process to make sure the image contains the correct angular light-field samples under each lenslet. The light field images are rectified to get 11 x 11 angular light-field samples (this number of angular points proved to work well and are used also in [6, 4]). It is used to make sure the perspective extraction is taking the pixels corresponding for a specific lenslet. If this is not done, the perspective extraction would not work, and would take shifted/blurred perspectives. We have provided more details on these operations in the methods sections.

25. What are the values on the colorbars on Supplementary Figs. 1b, c? It is very difficult to understand what we are looking at here and how to interpret colors. I would assume warmer colors on the SSIM mean better, and cooler colors for RSE. Please clarify.

Indeed, the referee is correct. We have now explicitly indicated this in the (new) main figures and supplementary figures.

26. The majority of the images in the SI are missing scale bars, and sometimes even when scale bars are given, there is no value in the caption (e.g., SI Fig. 2) Done
27. Scale bar values for SI videos? We apologize for having missed to label these.
28. In SI Video 2 (spheroid), there appear to be large fluctuations in the signal from one frame to the next. Is this because of focal drift? Please comment.
During imaging, spheroids were continuously perfused with fresh cofactor. Subtle variations in cofactor bioavailability and tissue distribution lead to transient intensity fluctuations.

References

- [1] Wang, H., et al., Deep learning enables cross-modality super-resolution in fluorescence microscopy. *Nature Methods* **16**, 103–110 (2019).
- [2] Ounkomol, C., Seshamani, S., Maleckar, M. M., Collman, F., Johnson, G. R., Label-free prediction of three-dimensional fluorescence images from transmitted-light microscopy. *Nature Methods* **15**, 917–920 (2018).
- [3] Merino, D., et al., STED imaging performance estimation by means of Fourier transform analysis. *Biomedical Optics Express* **8**, 2472 (2017).
- [4] Wang, Z., et al., Real-time volumetric reconstruction of biological dynamics with light-field microscopy and deep learning. *Nature Methods* **18**, 551–556 (2021).
- [5] Li, B., et al., Deep-3D microscope: 3D volumetric microscopy of thick scattering samples using a wide-field microscope and machine learning. *Biomedical Optics Express* **13**, 284 (2022).
- [6] Prevedel, R., et al., Simultaneous whole-animal 3D imaging of neuronal activity using light-field microscopy. *Nature Methods* **11**, 727–730 (2014).
- [7] Steinmeyer, R., et al., Improved fluorescent proteins for single-molecule research in molecular tracking and colocalization. *Journal of Fluorescence* **15**, 707–721 (2005).
- [8] Suzuki, K., et al., Five colour variants of bright luminescent protein for real-time multicolour bioimaging. *Nature Communications* **7**, 1–10 (2016).
- [9] Weigert, M., et al., Content-aware image restoration: pushing the limits of fluorescence microscopy. *Nature Methods* **15**, 1090–1097 (2018).

REVIEWERS' COMMENTS:

Reviewer #1 (Remarks to the Author):

The authors revised the manuscript accordingly. I do not have any further comments.

Reviewer #2 (Remarks to the Author):

I commend the authors on their thorough response – all my concerns have been addressed. I have the following minor edits:

Scale bars are missing on SI. Fig. 1b, c.

Fig. 1b caption: 'commercial epifluorescence (top) microscope...' I think the authors mean 'bottom', right?

Equation 5 (SNR) is missing a minus sign between the signal and background terms.

REVIEWERS' COMMENTS:

Reviewer #1 (Remarks to the Author):

The authors revised the manuscript accordingly. I do not have any further comments.

Reviewer #2 (Remarks to the Author):

I commend the authors on their thorough response – all my concerns have been addressed. I have the following minor edits:

Scale bars are missing on SI. Fig. 1b, c.

Fig. 1b caption: ‘commercial epifluorescence (top) microscope...’ I think the authors mean ‘bottom’, right?

Equation 5 (SNR) is missing a minus sign between the signal and background terms.

We thank the two referees for acknowledging our efforts to improve the presentation of the manuscript and foremost for their constructive comments. We have added the scalebar to figure S1 and eliminated the two typos.